**The Ross Sea and Amundsen Sea Ice-Sea Model (RAISE v1.0): a high-resolution ocean-sea ice-ice shelf coupling model for simulating the Dense Shelf Water and Antarctic Bottom Water in the Ross Sea, Antarctica**

5   Zhaoru Zhang[1,2,3,4], Chuan Xie[1], Chuning Wang[1], Yuanjie Chen[1], Heng Hu[1], Xiaoqiao Wang[1,5]

[1]Key Laboratory of Polar Ecosystem and Climate Change, Ministry of Education and School of Oceanography, Shanghai Jiao Tong University, 1954 Huashan Road, Shanghai, 200030, China

[2]Shanghai Key Laboratory of Polar Life and Environment Sciences, Shanghai Jiao Tong University, 10   Shanghai, China

[3]Shanghai Frontiers Science Center of Polar Science, Shanghai Jiao Tong University, 1954 Huashan Road, Shanghai, 200030, China

[4]Key Laboratory for Polar Science, Polar Research Institute of China, Ministry of Natural Resources, Shanghai, 200136, China

15   [5]High Impact Weather Key Laboratory of China Meteorological Administration (CMA), Changsha, 410073, China

*Correspondence to*: Zhaoru Zhang (zrzhang@sjtu.edu.cn)

**Abstract**

 The Ross Sea in the Southern Ocean is a key region for the formation of the Antarctic Bottom Water (AABW) that supplies the lower limb of the global overturning circulation, and contributes to 20–40% of the total AABW production. AABW primarily originates from polynyas characterized by strong sea ice production and ocean convection that lead to the formation of Dense Shelf Water (DSW), the precursor of the AABW. The production and characteristics of DSW in the Ross Sea and AABW in the surrounding ocean are significantly affected by ice shelf meltwater transported from the nearby Amundsen Sea. The scarcity of long-term observations in the Ross Sea hinders the understanding of DSW and AABW variability, and numerical models are needed to explore the multi-scale variations of these water masses and the forcing mechanisms. In this work, a coupled high-resolution ocean-sea ice-ice shelf model is developed for the Ross Sea and Amundsen Sea, named the **R**oss-**A**mundsen Sea **I**ce-**Se**a Model (RAISE). Detailed descriptions of the model configurations are provided. This study represents an attempt to thoroughly evaluate the DSW properties and associated ocean-sea ice-ice shelf coupling processes among modelling studies in the Southern Ocean, using multiple datasets including satellite-based observations and hydrographic measurements from the World Ocean Database, Argo profilers and seal-tag sensors. In particular, the modelled temporal variations of DSW properties in polynyas and its key export passages are compared with long-term mooring observations, which are rarely seen in studies of the DSW temporal variability before. RAISE demonstrates high skills in simulating the observed sea ice production rates in the Ross Sea polynyas, and the modelled spatial and temporal variability of DSW are significantly and strongly correlated with observations. RAISE can also effectively capture the observed long-term freshening trend of DSW prior to 2014 and the rebounding of DSW salinity after 2014. RAISE

shows an overestimate of DSW density in the Ross Sea, which is associated with an underestimate of ice shelf melting rates in the Amundsen Sea, missing ice shelf calving processes and subglacial discharge in the model. A sensitivity experiment simulating increased freshwater discharge from these processes can significantly improve the simulation of DSW properties.

45

# 1 Introduction

The Southern Ocean is the production site of bottom water mass in the global ocean — the Antarctic Bottom Water (AABW), which supplies the lower limb of the global thermohaline circulation. AABW primarily originates from polynyas on Antarctic continental shelves or in open ocean regions. These polynyas are characterized by low sea ice concentrations, facilitating substantial ocean-atmosphere heat fluxes during cold seasons, which drive significant new ice production. Brine rejection during ice formation further drives deep ocean convection, leading to the formation of the Dense Shelf Water (DSW) in the polynya regions. DSW is subsequently transported across the continental slope and into the open ocean, entraining other water masses such as the Ice Shelf Water (ISW) and the Circumpolar Deep Water (CDW), eventually contributing to the formation of the AABW.

The Ross Sea is a key region for the formation of DSW and contributes to 20–40% of the global AABW production (Meredith et al., 2013; Solodoch et al., 2022). DSW is primarily formed in two coastal polynyas, the Terra Nova Bay polynya (TNBP) off the Victoria Land and the Ross Ice Shelf polynya (RISP, also called the Ross Sea polynya) off the largest ice shelf in Antarctica (Fig. 1). Intense katabatic winds blowing from the Transantarctic Mountains or ice shelves toward the ocean drive the formation of these polynyas, enhancing sea ice production and the DSW formation. DSW is then transported to the slope along three deep troughs on the Ross Sea shelf— the Drygalski Trough, the Joides Trough and the Glomar Challenger Trough (Fig. 1). From there, it  sinks down to the ocean bottom or is carried to East Antarctic regions by the westward Antarctic Slope Current, forming AABW in the Pacific sector of the Southern Ocean. The formation and properties of DSW in the Ross Sea are significantly affected by freshwater input from melting ice shelves in the Amundsen Sea (Kusahara and Hasumi, 2013; Nakayama

et al., 2014). Under surface warming and enhanced on-shelf intrusion of the warm CDW, there has been accelerated melting of ice shelves in the Amundsen Sea, and the increased meltwater transport into the Ross Sea has caused a freshening trend in DSW over the past few decades (Jacobs et al., 2022). Such trend is found to be reversed in recent years due to interactions between major climate modes, leading to changes in winds and sea ice exchange between these marginal seas (Castagno et al., 2019; Silvano et al., 2020; Guo et al., 2020).

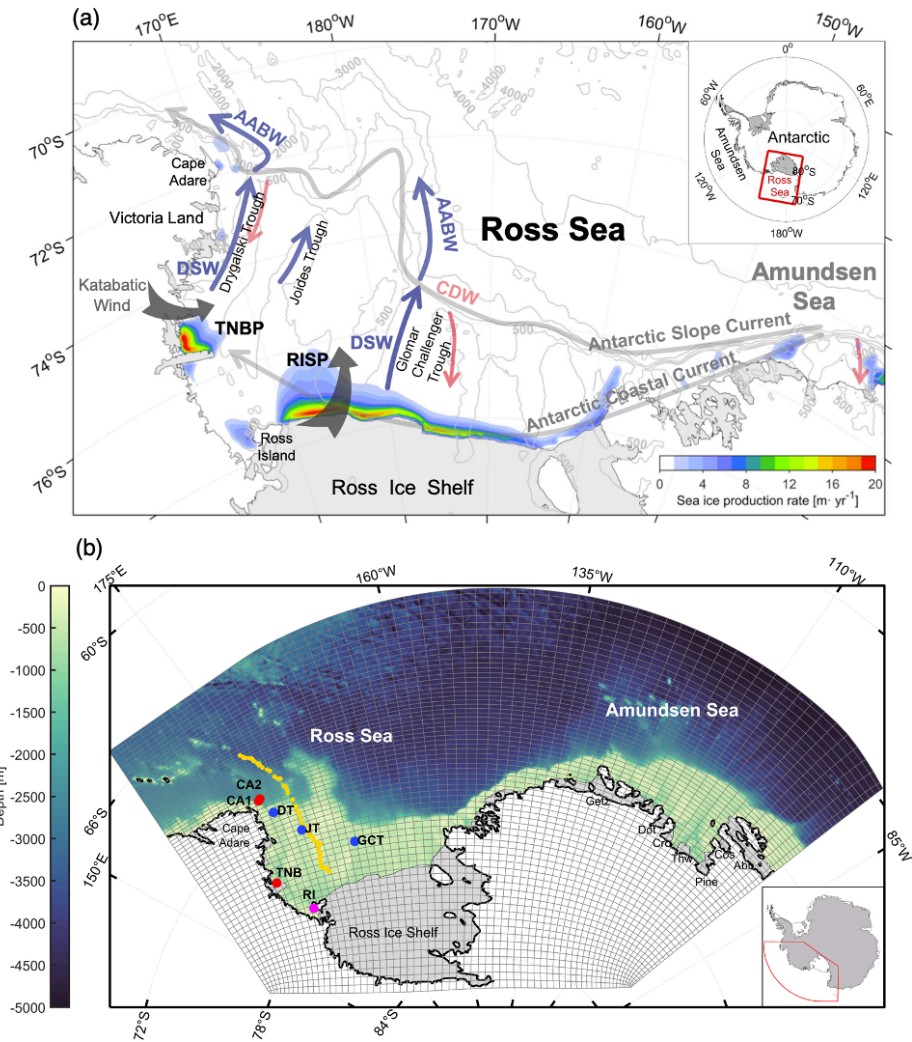

**Figure 1. (a)** Map of the Ross Sea and the Amundsen Sea. The bathymetric contours are shown as thin grey lines. Grey shading indicates ice shelves. The movement of the dense shelf water (DSW) and Antarctic Bottom Water (AABW) are illustrated by blue arrows, and the movement of the circumpolar deep water (CDW) is illustrated by red arrows. The Antarctic Slope Current and coastal current are illustrated by thick grey arrows. TNBP denotes the Terra Nova Bay polynya, and RISP denotes the Ross Ice Shelf polynya. Color indicates the climatological annual-accumulative sea ice production from satellite estimates based on the AMSR-E data by Nakata et al. (2021). **(b)** The RAISE model domain and grid. Grey color indicates ice shelves including the Ross Ice Shelf, Getz Ice Shelf (Getz), Dotson Ice Shelf (Dot), Crosson Ice Shelf (Cro), Thwaites Ice Shelf (Thw), Cosgrove Ice Shelf (Cos) and Pine Island Ice Shelf (Pine). The yellow line denotes the cross-shore transect from the MEOP dataset. The red dots indicate mooring locations in the Terra Nova Bay and on the Ross Sea slope near Cape Adare. The magenta dot indicates the long-term observational site near the Ross Island mentioned in Jacobs et al. (2022). The blue dots indicate locations for examining the long-term variations of DSW in the three troughs on the Ross Sea shelf.

In recent years, several ocean-sea ice-ice shelf coupling models have been developed for the Ross Sea. Dinniman et al. (2018) developed a 5-km coupled ocean-sea ice-ice shelf model for this area using the Regional Ocean Modelling System (ROMS), on the basis of an ocean circulation model (Dinniman et al., 2004) and a coupled ocean-ice shelf model (Dinniman et al., 2007; Dinniman et al., 2011). This model is employed to study the future changes of atmospheric forcings and freshwater inflow on the formation of DSW, the on-shelf intrusion of CDW and basal melting of the Ross Ice Shelf. In the model, changes in the freshwater inflow from the Amundsen Sea are simulated by reducing salinity at the eastern and western boundaries of the Ross Sea. Yan et al. (2023) developed a coupled ocean-sea ice-ice shelf model for the Ross Sea based on the Massachusetts Institute of Technology General Circulation Model (MITgcm), also with a horizontal resolution of approximately 5 km. This model is used to analyze the seasonality of salinity budget to understand the controlling mechanisms for the bottom water variation. Since DSW formation is significantly affected by the meltwater inflow from the Amundsen Sea ice

shelves, achieving realistic simulations of DSW characteristics requires incorporating processes in the
Amundsen Sea. This includes directly simulating the melting of ice shelves and assessing the impacts of
these meltwater on the Ross Sea shelf and slope environments.

Given the concerns above, in this work, we developed a high-resolution coupled ocean-sea ice-ice shelf model covering both the Ross Sea and the Amundsen Sea, named as RAISE (**R**oss-**A**mundsen Sea **I**ce-**Se**a Model). Compared to earlier modelling efforts, this study for the first time provides comprehensive evaluations of the performance of a coupled model in simulating the DSW properties, particularly its temporal variability, in the Ross Sea. These evaluations are conducted using cruise measurements, mooring observations and satellite-retrieved datasets. The model demonstrates good skill in capturing the variations of sea ice production rates in the Ross Sea polynyas, the long-term freshening trend of DSW, and the observed salinity rebound after 2014. Compared to mooring observations, the model also effectively captures DSW variations at higher frequencies at both the DSW formation sites and major export locations on the Ross Sea slope.

## 2 Model setup

### 2.1 The ocean model

The ocean model of RAISE is an implementation of the Regional Ocean Modelling System (ROMS), which is a primitive-equation, finite-volume model with a terrain-following vertical coordinate system (Haidvogel et al. 2008; Shchepetkin and McWilliams 2009). The model domain covers the Ross Sea, the Amundsen Sea and the adjacent open ocean in the Pacific sector of the Southern Ocean (Fig. 1b). This model is an updated version of the one used in Xie et al. (2024) and Zhang et al. (2024a), and the

major differences among these models are the application of nudging for temperature and salinity, as will be explained later in this section. The model horizontal resolution varies from ~2 km in the coastal areas to ~6 km in the open ocean. This model includes 32 vertical layers, with variable thicknesses that depend on water column depth and are smaller in the surface and bottom layers. On the Ross Sea shelf and slope, the thickness of the bottom layer varies from 10 m over banks to 60 m in the Drygalski Trough. In the open ocean, the bottom layer thickness varies from 100 m to 200 m. The model bathymetry and ice shelf draft are interpolated from BedMachine-Antarctica-v2.0 (Morlighem et al. 2020), which has a spatial resolution of 500 m on the Antarctic Polar Stereographic projection. Vertical mixing of momentum and tracers are computed using the K-profile parameterization (KPP) mixing scheme (Large et al. 1994).

Initial conditions of temperature and salinity come from simulations from a circum-Antarctic ocean-sea ice-ice shelf model with a horizontal resolution of 10 km (Dinniman et al. 2015). Alternative initial conditions from the World Ocean Atlas 2018 (WOA18) are also employed for this model, and we found that after a 5-year spin-up period, these conditions yield quite similar model simulations to those initialized by the model results from Dinniman et al. (2015). This indicates that the model simulations are not significantly affected by the initial fields. Temperature, salinity, sea surface height and depth-averaged velocity for the open boundaries are derived from daily data of the Met Office Global Seasonal forecasting system version 5 (Glosea5) (Maclachlan et al. 2015). Hydrographic simulations from five global ocean-sea ice reanalysis products are compared with the EN4 dataset (Good et al., 2013) for the Ross Sea, the Amundsen Sea and the nearby open ocean, including C-GLORS, GECCO3, GLORYS12V1, ORAS5 and Glosea5, and it is found that Glosea5 has the overall best performance in simulating temperature and salinity in this region (Fig. 2). Below 1000 m (the isobath at the shelf break), the average root mean square

errors of temperature and salinity over the model domain relative to the EN4 dataset are 0.165 °C and 0.054 psu, respectively. Tidal forcing is derived from the global tidal solution TPXO9 (Egbert and Erofeeva 2002), including 15 major tidal constituents (K1, S2, M4, P1, O1, Q1, S1, MS4, MN4, MF, 2N2, M2, K2, MM and N2) forced at the open boundaries via sea surface height and barotropic velocity.

Atmospheric forcing fields for the model are obtained from the ERA5 reanalysis product, including 3-hourly data for surface wind and air temperature, and daily data for sea level pressure, humidity, cloudiness and precipitation. Compared to the model used in Xie et al. (2024) and Zhang et al. (2024a), surface temperature and salinity are nudged to monthly mean climatology in this model, provided by the WOA18 dataset (Locarnini et al., 2018). Due to limited observational data in the Antarctic shelf regions

in WOA18, which are mostly collected during summer, surface nudging is only applied to the off-shelf regions. Such nudging results in improved simulations of sea ice production and DSW properties (Figs. S2 and S3 in the Supplementary Information).

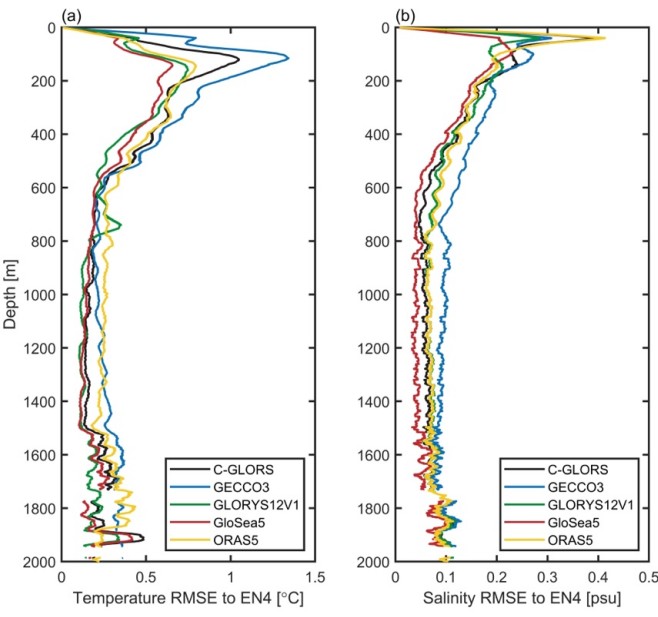

**Figure 2.** Vertical profiles of root mean square errors of **(a)** temperature and **(b)** salinity from C-GLORS, GECCO3, GLORYS12V1, GloSea5 and ORAS5 relative to the EN4 dataset.

Numerical dyes are released in the model to trace the movement and distributions of major water masses, including the DSW, CDW and ISW. The dyes are released continuously during the simulation periods of the experiments. DSW dyes are released at model grid points in the polynya areas where sea ice production occurs, and the dye values are proportional to the ice production rates. CDW dyes are released at grid points in the open ocean (offshore of the 1000-m isobath) where temperatures are greater than 0°C, with the initial dye values set as 100. ISW dyes are released at grid points where ice shelf exists, and the dye values are proportional to the ice shelf basal melting rates.

## 2.2 The sea ice module

The sea ice module (Budgell 2005) of RAISE is based on two-layer ice thermodynamics and a molecular sublayer beneath the sea ice described by Mellor and Kantha (1989) and Häkkinen and Mellor (1992), and elastic-viscous-plastic rheology for ice dynamics (Hunke and Dukowicz, 1997; Hunke, 2001). A snow layer is included, and snow is converted into ice when the snow-ice interface is below sea level. The sea-ice model also includes a simple estimate of frazil ice production (Steele et al. 1989). Boundary conditions of sea ice concentration are obtained from daily data of the Advanced Microwave Scanning Radiometer-Earth Observing System (AMSR-E) and the Advanced Microwave Scanning Radiometer 2 (AMSR2) datasets, provided by the University of Bremen using the ARTIST sea ice algorithm (Spreen et al. 2008).

## 2.3 The ice shelf module

The ice shelves in the model are static, and there are no thickness or extent changes of an ice shelf

over time. Configurations of the ice shelf module follow those in Dinniman et al. (2007) and Dinniman

et al. (2011). The hydrostatic pressure at the base of the ice shelf is computed based on the assumption

that ice is in isostatic equilibrium. Friction between the ice shelf and the water is computed as a quadratic

stress, and is applied as a body force over the top three ocean layers beneath the ice shelf. At the interface

between the ocean and ice shelf, a parameterization scheme with a viscous sublayer model is used with

three equations representing the conservation of heat across the ocean-ice shelf boundary, the

conservation of salt and a linearized version of the freezing point of sea water as a function of salinity

and pressure (Holland and Jenkins, 1999). The conservation of heat across the ocean-ice shelf boundary

is expressed as:

$$\frac{\rho_I w_B L_f}{C_{pw}} = -\rho_W \gamma_T (T_B - T_W),\tag{1}$$

where $\rho_I$ is the ice density and specified as 930 kg m$^{-3}$, $L_f$ is the latent heat of fusion of ice specified as

$3.34 \times 10^5$ J kg$^{-1}$, $C_{pw}$ is the specific heat capacity of seawater specified as 4000 J kg$^{-1}$ °C$^{-1}$, $\rho_W$ is the

seawater density of the uppermost ocean layer (kg m$^{-3}$), $w_B$ denotes the rate of ice melting (> 0) or

freezing (< 0) (m s$^{-1}$), $T_B$ is the interface temperature (freezing point), $T_W$ is the temperature of seawater

at a certain distance from the ocean-ice shelf interface, and $\gamma_T$ is the heat transfer coefficient (m s$^{-1}$)

representing the molecular and turbulent mixing coefficient of heat within the ocean boundary layer

adjacent to the ice shelf. The conservation of salt across the ocean-ice shelf boundary is expressed as:

$$\rho_I w_B S_B = \rho_W \gamma_S (S_B - S_W),\tag{2}$$

where $S_B$ is the salinity at the ocean-ice shelf interface, $S_W$ represents the salinity of the uppermost ocean

grid cell in the model and $\gamma_S$ is the salt transfer coefficient (m s$^{-1}$). $\gamma_T$ and $\gamma_S$ are specified following

McPhee et al. (1987) that assumes a viscous molecular sub-layer adjacent to the ice–ocean boundary. The last equation is a linearized version of the equation for the freezing point of sea water, which is written as:

$$T_B = aS_B + bP_B + c, \tag{3}$$

where the salinity coefficient $a$ is specified as $-5.7 \times 10^{-2}$ °C, the pressure coefficient $b$ is specified as $-7.61 \times 10^{-4}$ °C dbar$^{-1}$, and $c$ the depth of the ice shelf base. The variables $w_B$, $T_B$, and $S_B$ can be solved by simultaneously solving Equations 1–3.

Using the configurations described in Section 2.1–2.3, the model is integrated from 2003 to 2019 starting from a 5-year spin-up simulation, and the model simulation is referred to as the CTRL simulation. A sensitivity experiment Melt+ is conducted in which the basal melting rates of ice shelves in the Amundsen Sea are increased by tunning the heat and salt transfer coefficients ($\gamma_T$ and $\gamma_S$) in Equations 1 and 2 (see the details in Section 4.5), in order to explore the effects of underrepresented ice shelf melting and unrepresented ice shelf calving on freshwater fluxes and thus the DSW formation; detail information of this experiment is provided in Section 4.5.

## 3 Validation datasets and methodology

### 3.1 Ocean

For the validations of hydrographic properties and water masses including the DSW, we use hydrography data from the World Ocean Database (WOD) available at the National Centers for Environmental Information (https://www.ncei.noaa.gov/products/world-ocean-database) and Argo data provided by the International Argo Program and the national programs that contribute to it

(https://argo.ucsd.edu). Hydrographic measurements along a cross-shelf transect (Fig. 1b) from the Marine Mammals Exploring the Oceans Pole to Pole (MEOP) elephant-seal data (Treasure et al. 2017) are also used to assess the simulated water masses in the Ross Sea. Salinity data collected near the Ross

Island during summer (December to February) from Jacobs et al. (2022) are employed to evaluate the interannual variation and trend of the DSW salinity from 2003 to 2019. High-frequency variability of DSW formed in the Terra Nova Bay polynya is evaluated by hydrography measurements from a mooring (Fig. 1b) deployed by the MORSea Project of the Italian National Research Antarctic Program for 2008–2016. High-frequency variability of DSW on the slope is evaluated by measurements from two moorings

(Fig. 1b) deployed by the U.S. Cape Adare Long-term Mooring (CALM) program for 2008–2011.

### 3.2 Sea ice and ice shelf

Model simulations of sea ice concentration are compared with the AMSR-E and AMSR2 datasets provided by the University of Bremen, available as daily data with a horizontal resolution of 6.25 km (Spreen et al., 2008). The simulations of sea ice production (SIP) are evaluated against the satellite-

230 retrieved SIP dataset provided by the Institute of Low Temperature Science at the Hokkaido University (http://www.lowtem.hokudai.ac.jp/wwwod/polar-seaflux/southern_ocean_new/AMSR-POLAR/), which are calculated using the AMSR-E data for sea ice concentration and the ERA5 data for heat fluxes. This dataset includes estimates of frazil ice production (Nakata et al., 2019; Nakata et al., 2021). The data are provided monthly on the polar stereographic grid with a spatial resolution of 6.25 km, and are available

in the period of 2003–2010. For the assessment of temporal variations of SIP, we use a SIP product spanning a longer period, i.e. from 1992 to 2013 as described in Tamura et al. (2016). For this product, sea ice production is estimated by the heat flux calculation using the SSM/I passive microwave data and

atmospheric reanalysis products including ERA-40, ERA-interim and NCEP24; frazil ice is not included in this dataset. Melting rates of ice shelves in the Amundsen Sea are compared with the estimates from

Rignot et al. (2013), Depoorter et al. (2013) and Liu et al. (2015) based on satellite observations.

## 4 Simulation results

### 4.1 Sea ice concentration and production

The modelled spatial distributions of SIC during the ice freezing seasons are shown in Fig. 3,

assessed against SIC distributions derived from the AMSR-E/AMSR2 products. Compared with the satellite data (Fig. 3a–c), the RAISE model overall underestimates SIC over most of the model domain, which is attributed to high temperature bias in the surface ocean of the model as will be presented in Section 4.2. Correspondingly, the model underestimates the sea ice extent, but demonstrates good skill in capturing the temporal variability of the ice extent (Fig. 3g). Correlation between the modelled and

satellite-derived temporal variations of sea ice extent anomalies reaches 0.68 (P<0.001).

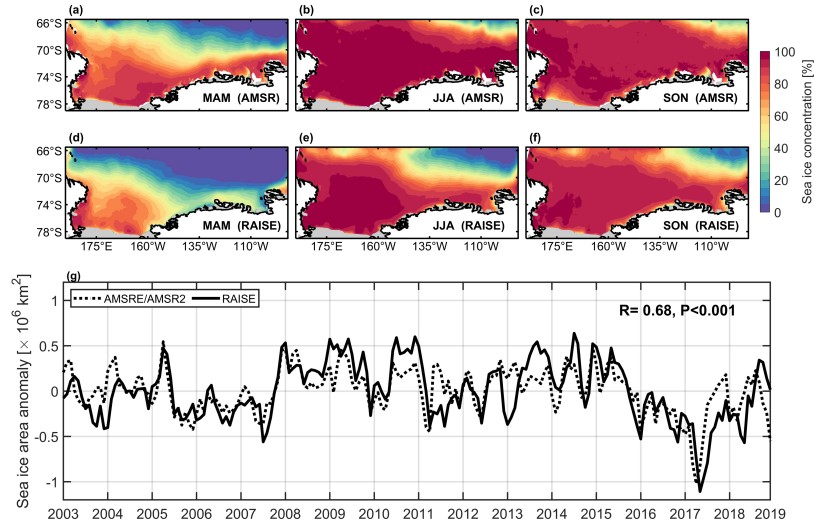

**Figure 3**. Seasonal mean sea ice concentration from **(a–c)** the AMSR product and **(d–f)** the RAISE model simulation for austral **(a, d)** autumn, **(b, e)** winter and **(c, f)** spring averaged over 2003–2019. **(g)** Time series of sea ice area anomalies from the AMSR-E/AMSR2 and from the RAISE simulation. The correlation coefficient (R) and p-value (P) between these time series are provided.

Sea ice production is the determinant factor for the DSW formation in the Southern Ocean. Comparisons between the modelled and satellite-estimated annual accumulative SIP rates (over the freezing seasons March to October) averaged over 2003–2010 are shown in Fig. 4. It can be seen that the model can well simulate locations and shape of the two major formation sites of DSW in the Ross Sea, the TNBP and RISP, while the simulated SIP rates are higher than satellite estimates by 14.9 km$^3$ and 236 km$^3$ for the TNBP and RISP area-mean values, respectively. Such differences are on the one hand due to inadequate representations of sea ice thermodynamic and dynamic processes in the model, which lead to an overestimate of sea ice thickness in the polynya areas (Fig. S1 in the Supplementary Information). On the other hand, there are also estimation errors in the satellite products. For example, satellite estimation does not include oceanic heat fluxes, and based on observed vertical temperature profiles in the Terra Nova Bay by Thompson et al. (2020), temperature in the subsurface layer in this region is lower than the surface layer, and hence there would be more sea ice production if the vertical oceanic heat fluxes are considered in the satellite retrieval algorithms. Interannual variations of the modelled annual accumulative SIP rates are significantly correlated with those from satellite estimates for both the TNBP (R=0.62, P=0.04) and RISP (R=0.55, P=0.08), demonstrating that the model can well simulate the temporal variability of ice production in the Ross Sea polynyas. Previous studies demonstrated that DSW primarily exists in the western portion of the RISP (Orsi and Wiederwhol, 2009; Wang et al., 2021), as the eastern portion receives more meltwater from the Amundsen Sea ice shelves and the Ross Ice Shelf.

Therefore, sea ice production in the western RISP contributes most to the DSW production in this polynya.
We compared the interannual variability of modelled sea ice production in the western RISP (west of 186°E) to that from the satellite estimate, and the correlation is significantly improved (R=0.76, P=0.01).

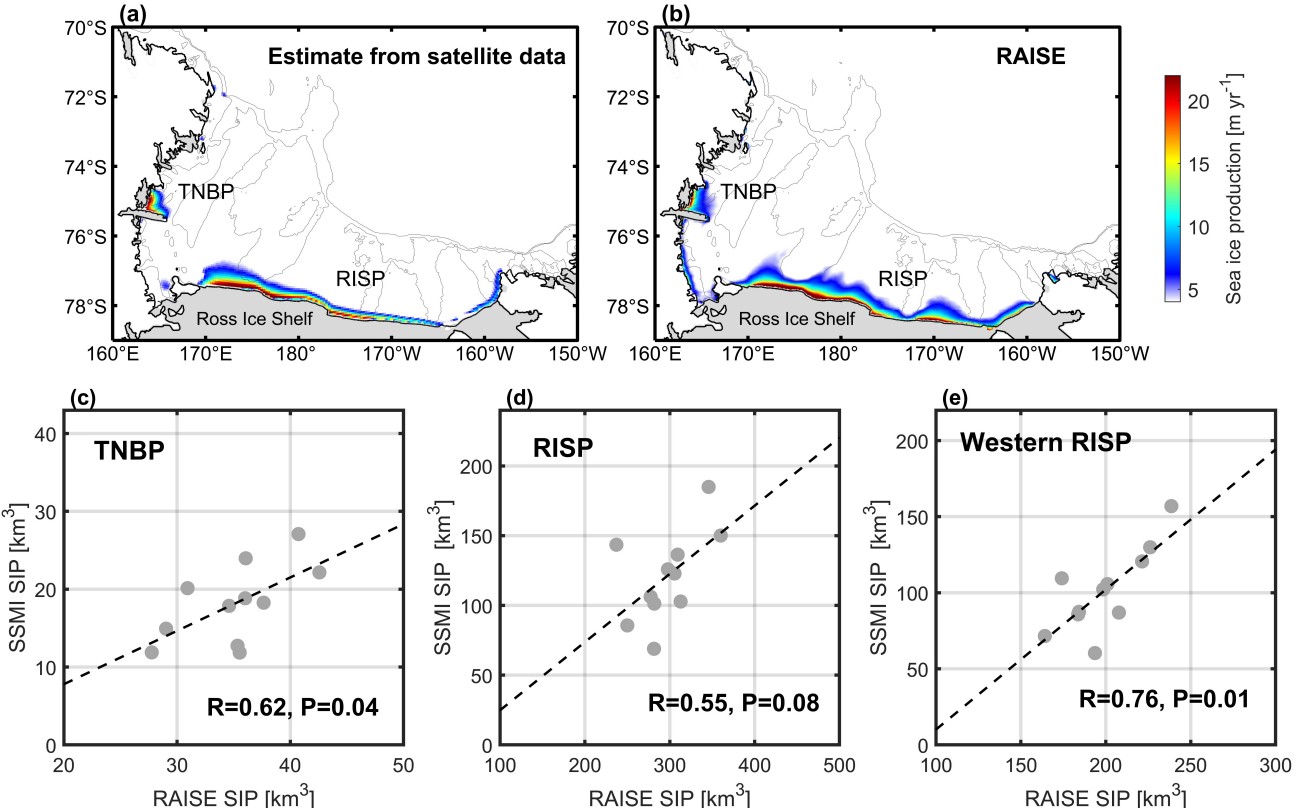

**Figure 4. (a)** Cumulative sea ice production in March–October averaged over 2003–2010 from satellite estimates. **(b)** The simulated cumulative sea ice production in March–October averaged over 2003–2019. **(c)** The scatter plot of cumulative sea ice production (unit: km$^3$) averaged over the Terra Nova Bay polynya from the model simulation versus that from satellite estimate. **(d)** Same as (c) but for the Ross Ice Shelf polynya. **(e)** Same as (d) but for the western Ross Ice Shelf polynya. The correlation coefficients (R) and p-values (P) between modelled and satellite-retrieved data are provided for (c), (d) and (e).

## 4.2 Hydrography and water masses

The temperature-salinity diagrams of Ross Sea waters below the depth of 100 m from WOD and

RAISE are shown in Fig. 5. The model can well depict the distributions of major water masses in the

subsurface and bottom layers of the Ross Sea, including the salty DSW, the warm CDW and the cold

ISW that all contribute to the formation of AABW. Compared to WOD, there is some deficiency for the

model to capture the high-salinity ends (34.9–35 g kg$^{-1}$) of DSW and high-temperature ends (1.5–1.7 °C)

of

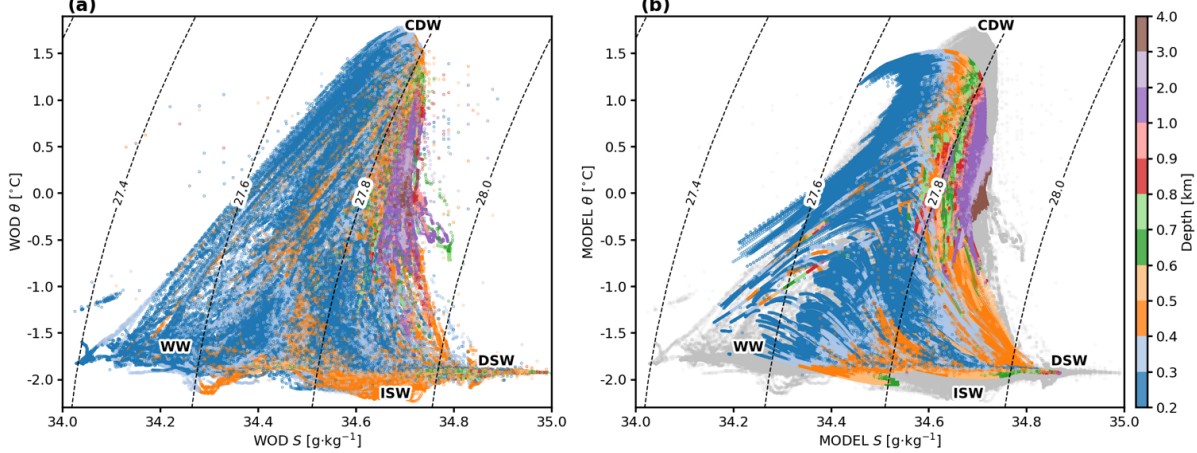

**Figure 5.** Temperature-salinity diagrams from **(a)** the WOD dataset and **(b)** the RAISE model simulation over the model domain. The major water masses are labeled, including the Dense Shelf Water (DSW), Circumpolar Deep Water (CDW), Ice Shelf Water (ISW) and Winter Water (WW).

CDW. Model simulations of potential temperature and salinity are also compared with seal-tag CTD

measurements from MEOP on a transect across the Ross Sea and the adjacent open ocean (Fig. 1b). The

observed spatial structures of temperature and salinity are well captured by the model (Fig. 6), and the

model also performs well in simulating the on-shelf intrusion of warm CDW and the distribution of dense

DSW (defined as neutral density $\gamma^n$ >28.27 kg m$^{-3}$). Compared to observations, the model slightly

overestimates temperature and underestimates salinity in the surface layer. In the subsurface layer, the model has lower temperature in the open ocean and higher temperature on the shelf, indicating stronger CDW intrusion in the model relative to the observational data; the subsurface salinity is underestimated in the model. The warm bias in the surface layer of the model is responsible for the underestimation of sea ice concentration in the model as shown in Fig. 3.

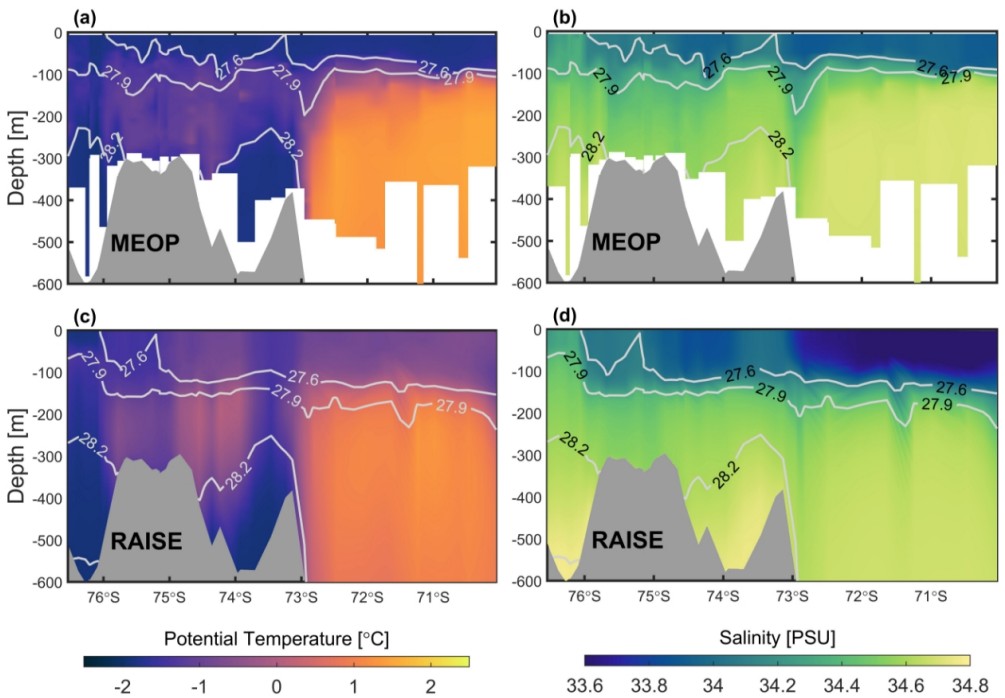

**Figure 6.** Vertical sections of **(a, c)** potential temperature and **(b, d)** salinity from **(a, b)** the MEOP data and **(c, d)** the RAISE model simulation along the cross-shore transect in the Ross Sea and open ocean (shown in Fig. 1b). Contours indicate isolines of neutral density.

The climatological (2003–2019 average) model simulations of potential temperature and salinity in the bottom 100-m layer (the layer mainly composed of DSW or AABW) on the Ross Sea shelf and slope and the adjacent open ocean are evaluated against climatology from the WOD and Argo data (Fig.

7). The spatial distributions of modelled temperature and salinity compare well with those from observations (Fig. 7a, c). Linear regression results reveal that the spatial correlation between the modelled and observed potential temperature reach 0.91 (Fig. 7b), with a slope value of 0.86. The correlation between modelled and observed salinity is 0.80 (Fig. 7d), with a slope value slightly lower than that for temperature and approaching 0.7. These results suggest that the climatological spatial patterns of DSW hydrography in the Ross Sea and AABW hydrography in the open ocean can be well represented by the model. It is noted that the model overestimates salinity in the lower-salinity range and underestimates salinity in the higher-salinity range (Fig. 7d). This means while the model produces larger salinity in the polynyas compared to observations, in other regions featured by high salinities there can be underestimates of salinity.

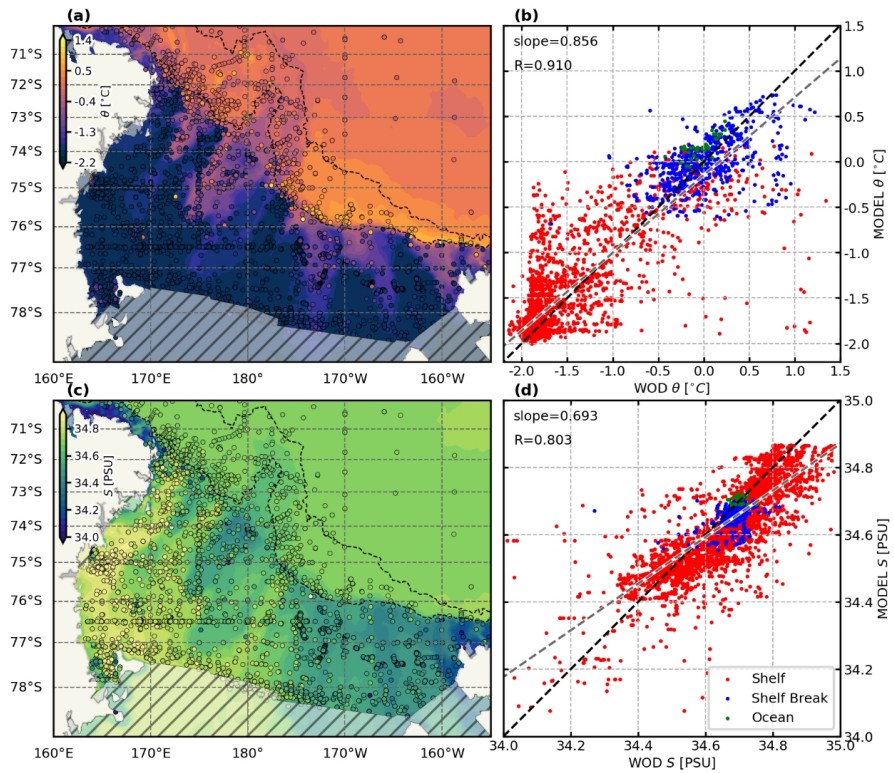

**Figure 7.** Comparisons of modelled climatological fields with historical hydrographic observations. **(a, c)** Modelled climatological potential temperature and salinity fields in the bottom 100-m layer, overlaid with historical observations (circled points). Historical data include all CTD, XBT, MBT, drifting buoy, glider and ocean station profiling measurements from the WOD and Argo data. **(b, d)** Scatter plots of modelled and observed potential temperature and salinity in the Ross Sea. Black dashed lines denote 1:1 ratio lines, and gray dashed lines denote linear regression fits. Colors of points denote the location of measurements; the three regions (shelf/shelf break/ocean) are separated by the 700-m and 3000-m isolines on the shelf break (marked by dashed lines in a, c).

The intrusion of warm CDW is a major mechanism for causing ice shelf basal melting and generating the ISW, which subsequently affects the DSW characteristics. The CDW dyes are initially released at model grid points in the open ocean where water temperatures are above 0°C. As seen in Fig. 8a that shows the CDW dye values 5 years after release at the 15$^{th}$ model level (200–400 m), CDW mainly intrudes onto the continental shelves via troughs and spreads over the shelf regions. High dye values are also present beneath the Ross Ice Shelf north of 80°S in the eastern Ross Sea, while low dye values reach much further south (to 82°S) beneath the ice shelf in the western Ross Sea, which could result from a southward flow that has been reported in earlier studies (Budillon et al., 2003; Jendersie et al., 2018; Stewart et al., 2019). The presence of CDW under the ice shelf is essential for its basal melting rate. Figure 8b and 8c show the distributions of vertically integrated values of ISW dye originating from ice shelves in the Ross Sea and Amundsen Sea, respectively. Compared to the western portion of the RIS, there is more ISW beneath the eastern portion, indicating stronger influence of ISW from this area on the the Ross Sea shelf hydrography. The ISW dye values are much higher in the Amundsen Sea and decrease dramatically towards the Ross Sea, indicating strong basal melting of ice shelves in the former region that provides fresh meltwater input to the Ross Sea, which plays a more important role in modulating the

salinity and stratification on the Ross Sea shelf compared to the meltwater released from the RIS. The transport time of ISW dyes from the Amundsen Sea to the Ross Sea is about 2 years.

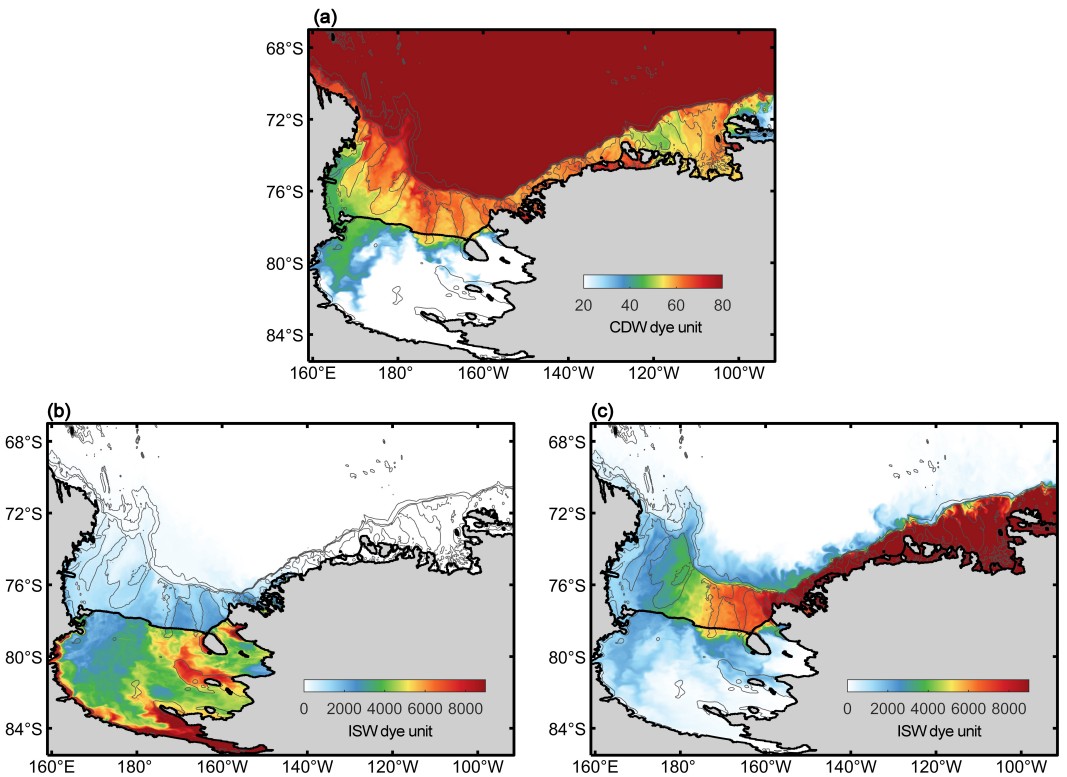

**Figure 8.** Values of **(a)** CDW dyes, **(b)** ISW dyes originating from the Ross Ice Shelf, and **(c)** ISW dyes originating from the Amundsen Sea ice shelves 5 years after the release time of the dyes in the model simulation.

In Fig. 9, DSW dyes are released separately for the TNBP and RISP. We can see that DSW formed in the TNBP (Fig. 9a) is mainly transported to the slope via the Drygalski Trough, and DSW formed in the RISP (Fig. 9b) is transported mainly via the Joides Trough and the Glamor Challenger Trough, while a portion also flows to the slope via the Drygalski Trough. Exports of DSW through the Drygalski Trough, Joides Trough and Clamor Challenger Trough contribute to the total DSW export by 41%, 14% and 45%,

respectively. Once crossing the slope and reaching the deep ocean, DSW turns into AABW and is mainly transported westward toward the Indian sector of the Southern Ocean by the Antarctic Slope Current.

DSW dyes released in the RISP cover a larger area in the open ocean than those released in the TNBP. DSW formed in the western portion of RISP is also carried southward beneath the Ross Ice Shelf, and can reach as far as 84°S near the grounding line of the ice shelf, which can be associated with the southward flow as mentioned above as well as the role of tidal currents (Arzeno et al., 2014). Such intrusion can be important for the basal melting of the Ross Ice Shelf, which is categorized as "cold-water

cavity" where the DSW acts as the main thermal forcing (Rignot et al., 2013; Adusumilli et al., 2020).

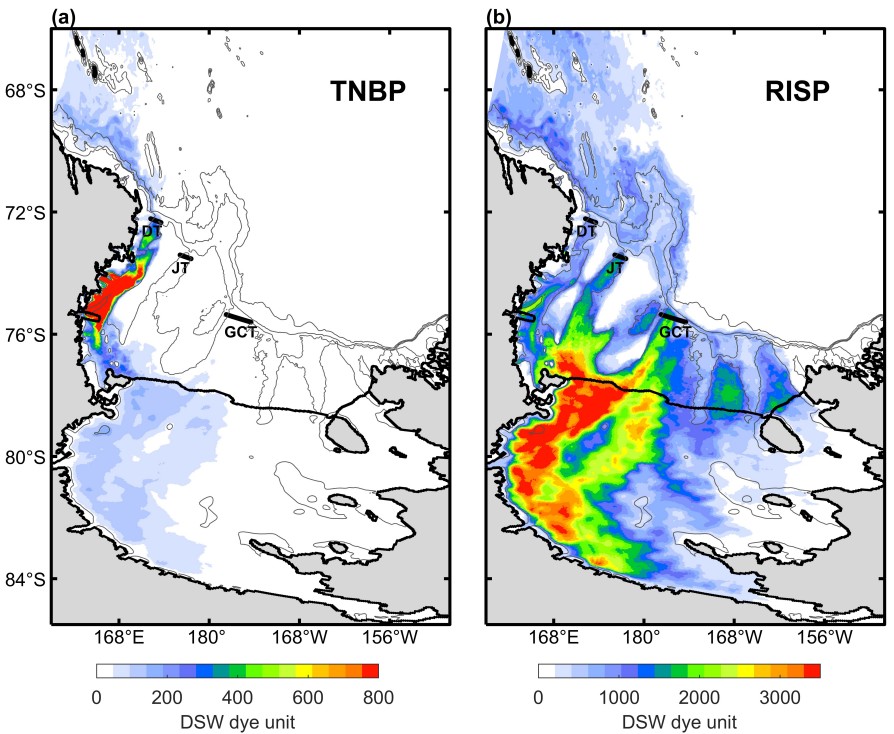

**Figure 9.** Vertically integrated values of DSW dyes originating from the **(a)** Terra Nova Bay polynya (TNBP) and **(b)** the Ross Ice Shelf polynya (RISP) 5 years after the dye release time in the model simulation. DT, JT and GCT denote the Drygalski Trough, Joides Trough and Glamor Challenger Trough, respectively.

## 4.3 Temporal variability of DSW

Temporal variations of neutral density in the middle and bottom layers of the TNBP are compared to measurements from mooring observations conducted by the Italian MORSea project (Fig. 10). Both the variations of neutral density at middle (500 m) and bottom depths (1060 m) are examined. While the

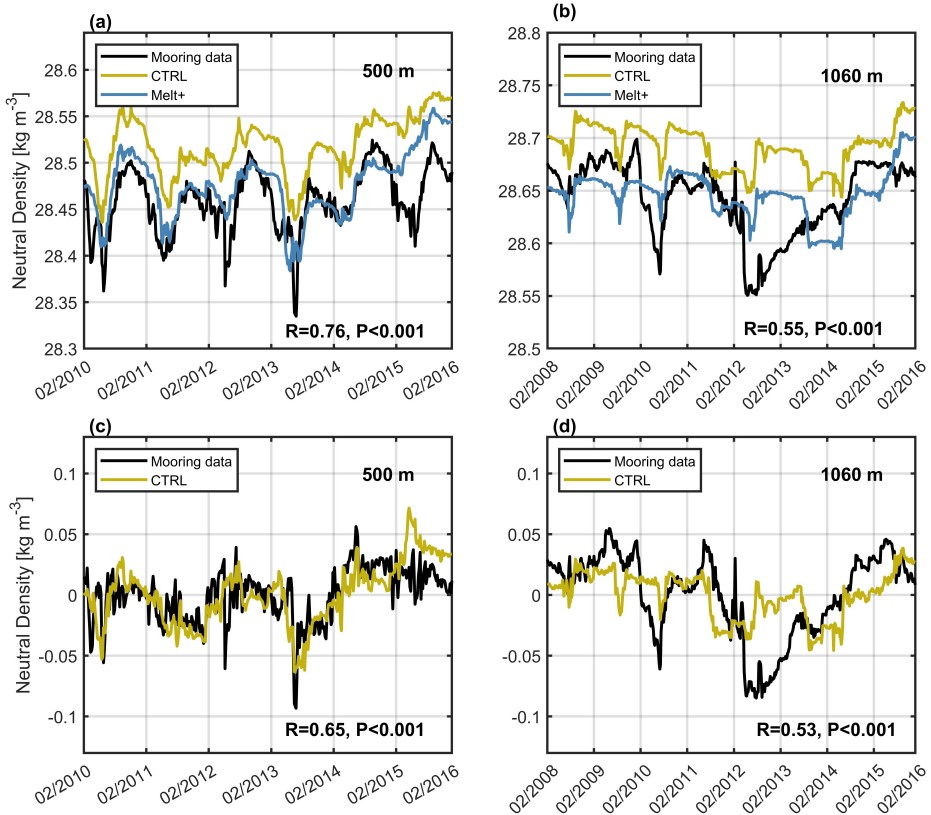

**Figure 10. (a)** Time series of 5-day-average neutral density at 500 m from the CTRL simulation, the Melt+ simulation and mooring observations in the Terra Nova Bay polynya (TNBP, see the mooring location in Fig. 1b) during 2010–2016. **(b)** Same as (a) but for neutral density at 1060 m during 2008–2016. **(c)** Time series of neutral density anomalies at 500 m from the CTRL simulation and mooring observations in the TNBP during 2010–2016. **(d)** Same as (c) but for neutral density at 1060 m during 2008–2016. The coefficient of correlation (R) between neutral density (anomaly) from the CTRL simulation and mooring and the corresponding p-value (P) are provided for each panel.

model has an overestimate of density in both layers compared to the mooring observations, which might be related to the model overestimate of sea ice production and inadequate representations of ice shelf melting processes (see the discussions in Section 4.4), the model can well capture the temporal variability of the DSW density. Correlations between the variations of modelled and observed neutral density at middle and bottom depths reach 0.76 and 0.55, respectively, both of which are significant (P<0.001). Removing the seasonal cycles, correlations between the variations of modelled and observed neutral density anomalies at middle and bottom depths are 0.65 and 0.53, respectively (P<0.001). This demonstrates the model can reasonably simulate the temporal variations of ocean-sea ice processes forming DSW at its originating sites. The annual average DSW production rate in the TNBP estimated from the RAISE model simulation is 0.33 Sv, which is between the estimate of 0.28 Sv from Jendersie et al. (2018) using a coupled ocean-sea ice model for the Ross Sea and the estimate of 0.43 Sv from Miller et al. (2024) using observations from a mooring in the TNBP. The estimated annual average DSW production rate in the RISP from the model is 1.23 Sv. The model also performs well in simulating the temporal variations of DSW properties at its key outflow site on the slope, as shown in Fig. 11. Comparisons with observations from two moorings deployed by the U.S. CALM project near Cape Adare (CA1 and CA2 in Fig. 1b) show that correlations between the simulated and observed variations of DSW neutral density near the ocean bottom reach 0.75 and above (P<0.001). The model also overestimates the density, possibly associated with the model bias in the DSW density in the polynya areas. Removing the seasonal cycles, correlations between the simulated and observed variations of DSW neutral density reach 0.70 at 1735 m and 0.63 at 1929 m, both of which are significant (P<0.001). Time series of the estimated DSW outflow fluxes at the exits of the three troughs (across the transects shown in Fig. 9) for different

years in 2003–2019 and for the multi-year average are provided in Fig. 12. For the Drygalski Trough, the Joides Trough and the Glamor Challenger Trough, the transports are relatively strong during February–

May, April–July and September–December, respectively; the annual mean outflow fluxes for the three troughs are 0.62 Sv, 0.19 Sv and 0.81 Sv, respectively.

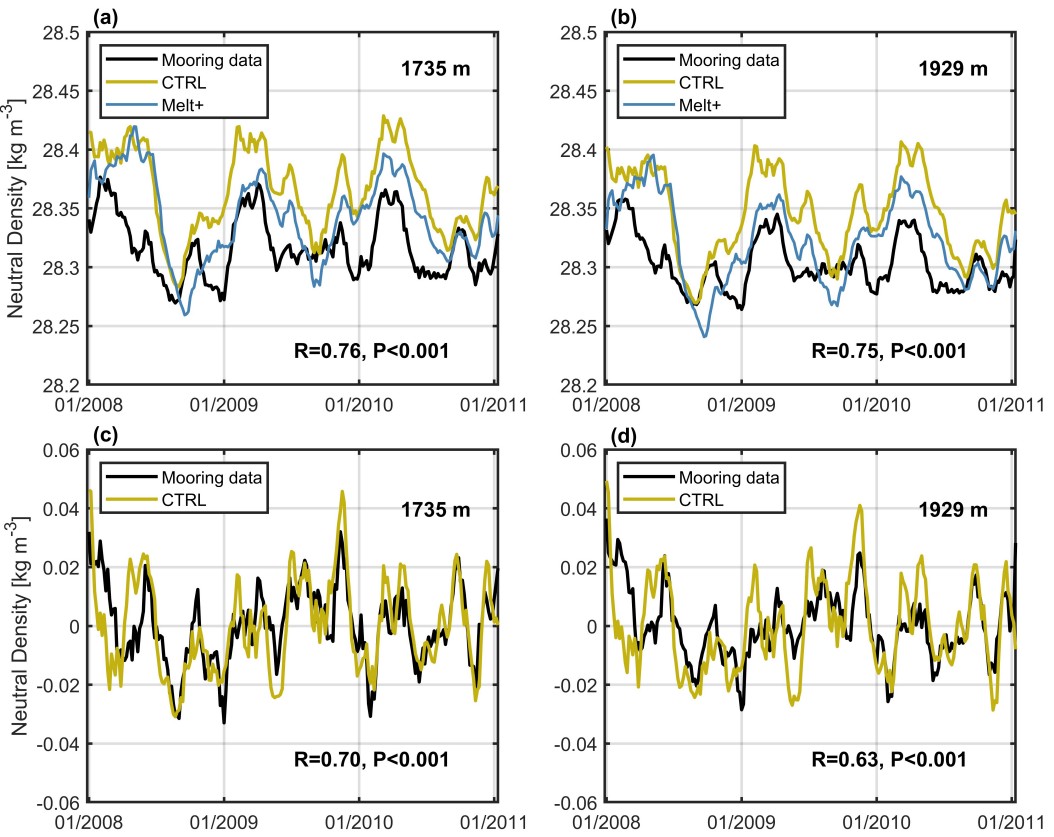

**Figure 11. (a)** Time series of 5-day-average neutral density at 1735 m from the CTRL simulation, the Melt+ simulation, and the CA1 mooring observations at the slope near Cape Adare (see Fig. 1b) during 2008–2011. **(b)**
Same as (a) but for neutral density at 1929 m, and the observations are from the CA2 mooring. **(c)** Time series of neutral density anomalies at 1735 m from the CTRL simulation and mooring observations at CA1 during 2008–2011. **(d)** Same as (c) but for neutral density at 1929 m at the CA2 mooring location during 2008–2011. The coefficient of correlation (R) between neutral density from the CTRL simulation and mooring, along with the corresponding p-value (P), are provided for each panel.

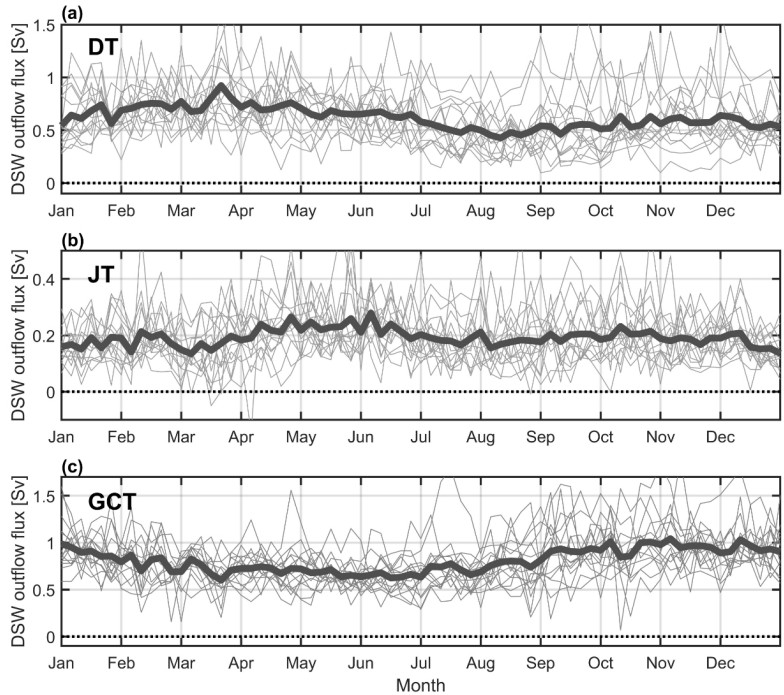

**Figure 12.** Time series of the DSW outflow fluxes at the exits of **(a)** the Drygalski Trough, **(b)** the Joides Trough and **(c)** the Glamor Challenger Trough for different years in 2003–2019 (grey thin lines) and the multi-year average (black thick line).

Over the past ten years, studies on DSW have focused on its decadal variation, revealing a
freshening trend in the Ross Sea based on observations (Jacobs et al., 2010; Jacobs et al., 2022). This
trend is attributed to increased transport of ice shelf meltwater from the Amundsen Sea into the Ross Sea
(Nakayama et al, 2014). Recent work found that such trend is reversed since 2014 (Castagno et al., 2019),
attributing the reversal to the combined effects of positive phase of the Southern Annular Mode and
extreme El Niño conditions (Silvano et al., 2020), and reduced input of meltwater from the Amundsen
Sea (Guo et al., 2020). Comparing the model simulations of DSW salinity from 2003 to 2019 with
observational data near the Ross Island from Jacobs et al. (2022), we found that the RAISE model
effectively captures both the freshening trend prior to 2014 and its reversal after 2014 (Fig. 13a). The

interannual variations of modelled and observed DSW salinity are significantly correlated (R=0.66, and P=0.004). The decadal variations of bottom water salinity are also examined for four locations in the three

troughs for DSW exports and the TNBP, which all show freshening trend of DSW before 2014 and rebounding of salinity after 2014 (Fig. 13b–e). The estimated freshening trend prior to 2014 varies from -0.008 PSU yr$^{-1}$ to -0.004 PSU yr$^{-1}$ at the four locations, which falls in the range of -0.08 to -0.01 PSU per decade as estimated by Castagno et al. (2019) based on long-term cruise observations.

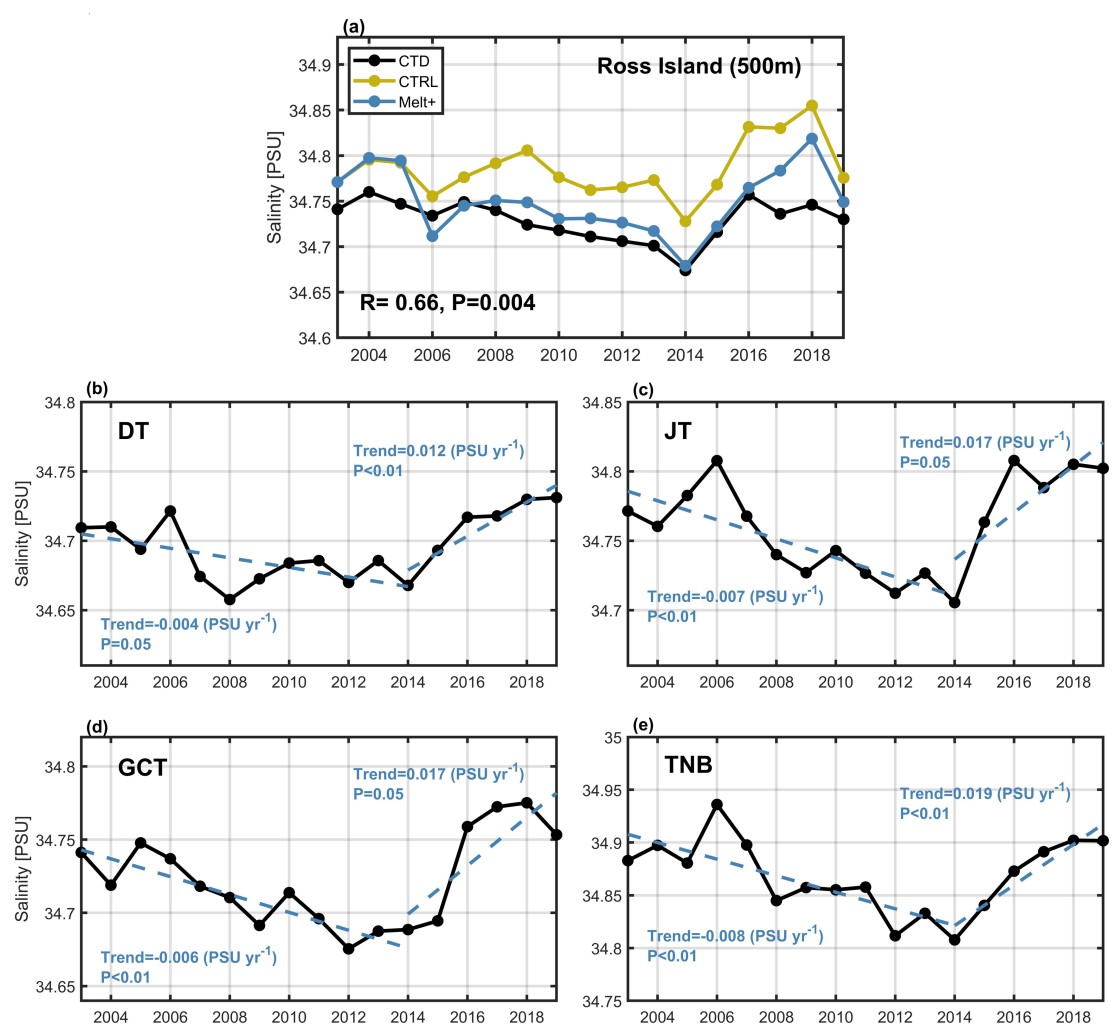

**Figure 13. (a)** Time series of summer bottom water salinity near the Ross Island from the CTRL simulation, the Melt+ simulation and CTD observations from Jacobs et al. (2022) during 2003–2019. **(b–e)** Time series of simulated summer bottom water salinity in CTRL at the long-term observation locations in **(b)** the Drygalski Trough, **(c)** the Joides Tough, **(d)** the Glamor Challenger Trough and **(e)** the Terra Nova Bay during 2003–2019. See the locations in Fig. 1b. The trends for the periods prior to 2014 and after 2014 with significance test results are labeled in panels b–e.

## 4.4 Ice shelf melting rates

As the melting of ice shelves in the Amundsen Sea and the Ross Sea can have significant impacts on the salinity and stratification in the Ross Sea, and thus the formation of DSW, the simulated melting rates of ice shelves are evaluated against estimates based on satellite data (Fig. 14). In the Ross Sea, the modelled melting rate for the Ross Ice Shelf averaged over 2003–2019 is about 79 Gt yr$^{-1}$, which is in line with 47.7±34 Gt yr$^{-1}$ estimated by Rignot et al. (2013) from remote sensing, while higher than the estimates of 34±25 Gt yr$^{-1}$ by Depoorter et al. (2013) and 27±22 Gt yr$^{-1}$ by Liu et al. (2015). In the Amundsen Sea, the simulated melting rate of the Getz Ice Shelf is higher than the satellite estimates from the studies mentioned above by 58–71 Gt yr$^{-1}$, while for the Dotson, Crosson, Thwaites and Pine Island ice shelves, the simulated melting rates are significantly lower than all satellite estimates. In total, in the Amundsen Sea the RAISE model underestimates the ice shelf melting rates by 107–172 Gt yr$^{-1}$ compared with the satellite-retrieved values. Such underestimates are largely attributed to the absence of subglacial runoff in the RAISE model, which is demonstrated to impose dramatic effects on basal melting of Antarctic ice shelves (Nakayama et al., 2021; Goldberg et al., 2023; Gwyther et al., 2023). In addition to underestimating ice shelf melting rates, the model does not account for ice shelf calving due to the use of a static ice-shelf module. Ice shelf calving is not included in the satellite estimates by Rignot et al. (2013)

and Depoorter et al. (2013) as well, and is separately considered in Liu et al. (2015) apart from the basal

melting process. Liu et al. (2015) suggests that ice calving can contribute to a mass loss of ice shelves of

270±22 Gt yr$^{-1}$ in the Amundsen Sea. The inadequate representation of freshwater input, due to

underestimated basal melting rates and the absence of ice shelf calving processes in the Amundsen Sea

could have significant influence on freshwater volume in the Ross Sea. This occurs through the westward

transport of meltwater by the Antarctic Slope Current and coastal currents and may contribute to the

model overestimate of DSW salinity in the Ross Sea. In the next section, a sensitivity experiment is

conducted to evaluate the effects of missing freshwater discharge associated with these processes on the

Ross Sea DSW characteristics.

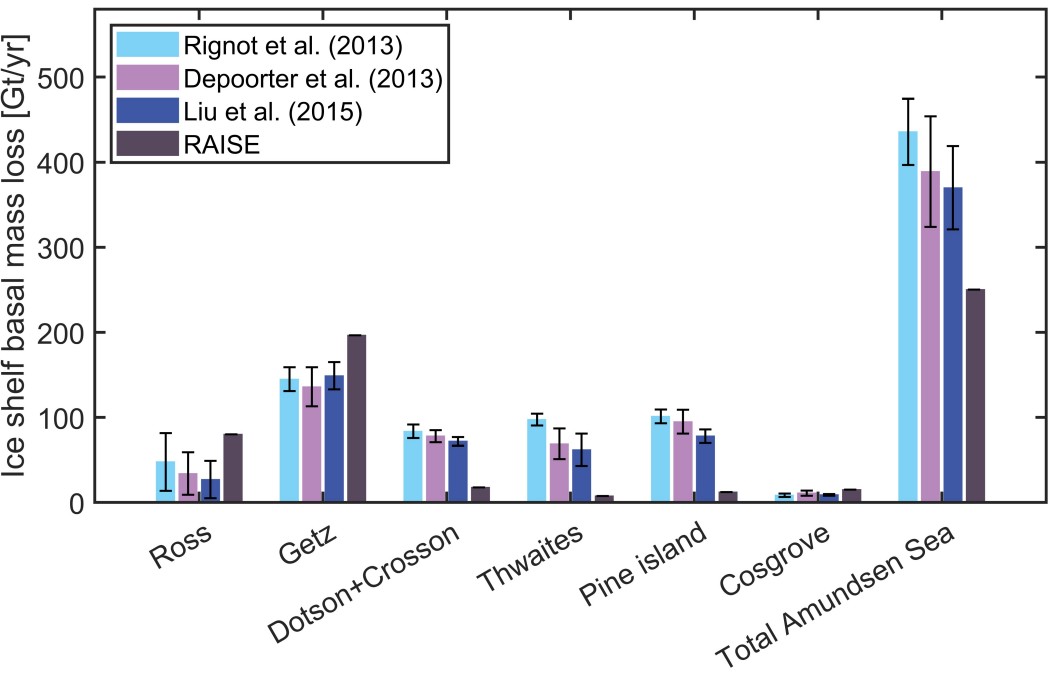

**Figure 14.** Basal melting rates of ice shelves in the Ross Sea and Amundsen Sea from the RAISE simulation and satellite estimates from earlier studies.

**4.5 A sensitivity experiment of increasing freshwater discharge associated with ice shelf basal melting and calving**

As shown in Figs. 10 and 11, the RAISE simulation shows overestimate of DSW neutral density compared to mooring observations in the Ross Sea polynya and slope regions. A possible reason for such bias is the inadequate representation of freshwater input from underrepresented basal melting rates and missing calving processes of ice shelves in the RAISE model, as discussed in Section 4.4. To test the role of these processes on the Ross Sea DSW properties, we conducted a sensitivity experiment Melt+, in which the basal melting rates of ice shelves in the Amundsen Sea are artificially increased to compensate for the missing freshwater discharge associated with the underestimated melting rates, absence of ice shelf calving as well as subglacial runoff. Contributions of these processes to the freshwater discharge sum up to ~450 Gt yr$^{-1}$ based on the estimates in Section 4.4, including the contribution from subglacial runoff in the Amundsen Sea estimated as ~10 Gt yr$^{-1}$ by Goldberg et al. (2023). Increases of the basal melting rates are achieved by modulating the heat and salt transfer coefficients at the ocean-ice shelf interface following Nakayama et al. (2020). The Melt+ experiment spans the period of 2003 to 2019. Time series of the DSW neutral density in the TNBP and on the Ross Sea slope in Melt+ are presented in Fig. 10 and Fig. 11, respectively. These values are substantially lower than those from the CTRL simulation and much closer to the observations. Compared to the CTRL simulation, in Melt+, the root-mean-square error of the DSW neutral density is reduced by 0.028 kg m$^{-3}$ in the TNBP, and reduced by 0.019 kg m$^{-3}$ on the Ross Sea slope. The salinity in the area near the Ross Island is also notably reduced (by 0.032 PSU) in Melt+ compared to that from CTRL (Fig. 13a). There is substantial decrease of salinity over the Ross Sea shelf by 0.02–0.1 PSU (Fig. 15), followed by a reduction in DSW thickness on the Ross Sea shelf and slope,

reaching up to 100 m, and a decrease of the AABW thicknesses across most of the open ocean by over 100 m. These results suggest that accurate representation of ice shelf melting, calving and subglacial runoff (note that it can also cause ice shelf melting) processes are crucial for accurate simulations of DSW formation and properties in the Ross Sea, which will then affect the simulation of AABW production in

the open ocean.

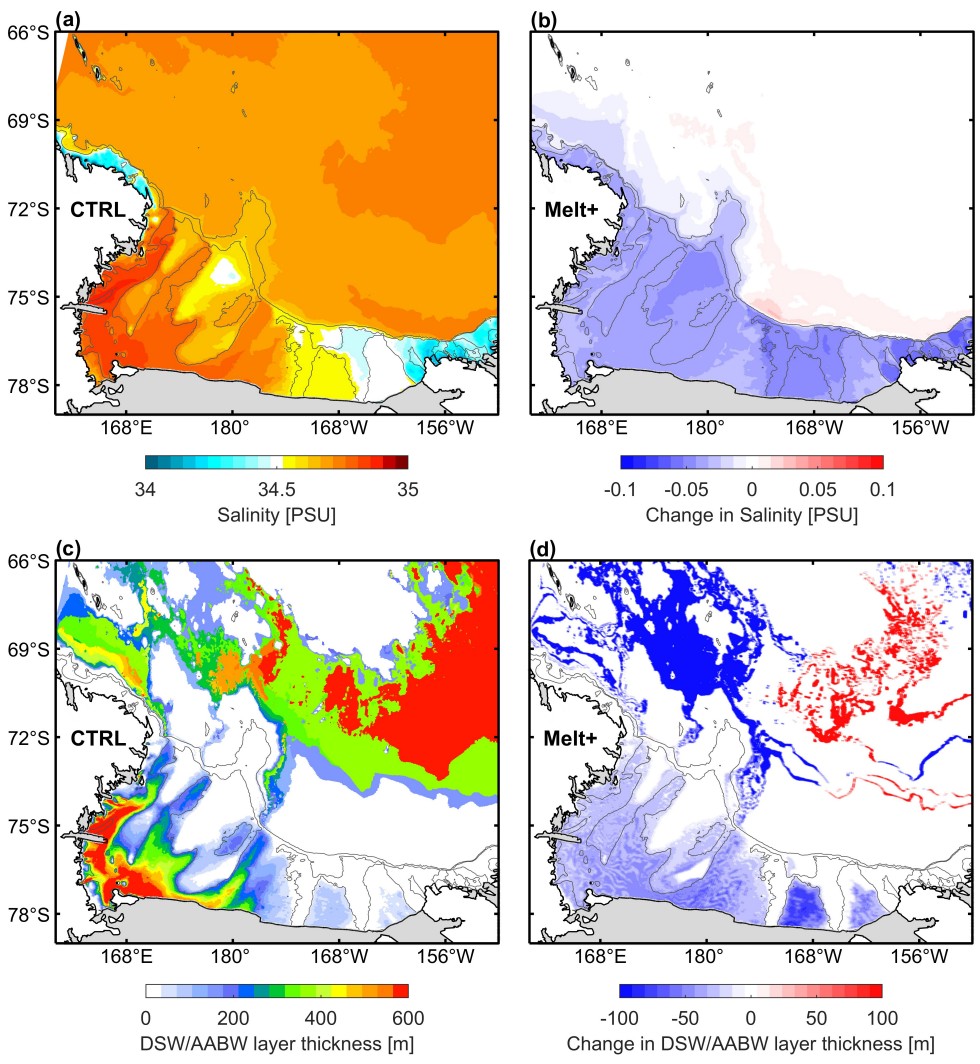

**Figure 15. (a)** Spatial distributions of salinity in the model bottom layer of the Ross Sea averaged over the simulation period. **(b)** Changes of salinity in the model bottom layer in Melt+ relative to CTRL. **(c)** Spatial distributions of DSW thicknesses on the Ross Sea shelf and slope and AABW thicknesses in the open ocean in the CTRL simulation. **(d)** Changes of DSW and AABW thicknesses in the Melt+ simulation relative to the CTRL simulation.

## 5 Conclusions and prospects

In this work, a high-resolution coupled ocean-sea ice-ice shelf model (RAISE) is developed for the Ross Sea and Amundsen Sea in the Southern Ocean. A major function of this model is to simulate the formation of Dense Shelf Water in the Ross Sea and the Antarctic Bottom Water in the open ocean in the Pacific sector, which is controlled by sea ice production in coastal polynyas and on the continental shelf, along with the discharge of freshwater from ice shelf melting, which is further influenced by the intrusion of the warm Circumpolar Deep Water. The RAISE model effectively simulates the spatial distributions and temporal variations of sea ice production rates, aligning well with satellite estimates. The modelled temperature and salinity distributions of DSW in the Ross Sea show good agreement with observations from the combined WOD and Argo data. The simulated temporal variations of DSW hydrography in both the Terra Nova Bay polynya and slope region of the Ross Sea are significantly and highly correlated with those obtained from mooring measurements. The RAISE model can also well capture the freshening trend of DSW prior to 2014 and the salinity rebounding after 2014. Compared with satellite estimates, the RAISE model significantly underestimates the melting rates of ice shelves in the Amundsen Sea, which is an important reason for the overestimate of DSW density in the Ross Sea. In a sensitivity experiment in which the basal melting rates of ice shelves are increased to compensate for the underrepresented ice shelf melting rates and the absence of ice shelf calving and subglacial runoff processes, the DSW density

is notably reduced compared to the CTRL simulation and is in better agreement with observations. Such results demonstrate the importance of accurate representation of freshwater released from ice shelves for accurately simulating the DSW formation and hydrography.

In the future, the model configurations can be further optimized to improve the simulations for DSW. First, the modelled sea ice production rates in the polynyas are higher than satellite estimates. While satellite products cannot be treated as ground truth as a couple of physical processes are missing in the retrieval algorithms, such as oceanic heat fluxes, the model may also have incomplete or misrepresented processes that lead to an overestimate of sea ice production. The ice-ocean drag or ice-atmosphere drag parameterization schemes can be tuned to yield better simulations of sea ice production. Second, the RAISE model resolution is about 5 km in the slope area, coarser than the baroclinic Rossby deformation radius in this region that is suggested to be ~1 km (Mack et al. 2019; Stewart and Thompson 2015). This will result in inadequate representation of CDW on-shelf intrusion driven by eddy activities, and may be one of the reasons for inadequate representation of ice shelf basal melting rates in the model. The model horizontal resolution can be further enhanced to improve the CDW intrusion simulation. Moreover, subglacial runoff is not included in the RAISE model, and such runoff can on the one hand contribute directly to the freshwater fluxes, and on the other hand contribute to ice shelf melting and consequently freshwater discharge. Subglacial discharge needs to be included in the model for more accurate simulation of freshwater volume and stratification in the Ross Sea. Finally, the RAISE model uses a static ice shelf module, and in the future a dynamic ice shelf module should be considered to include the ice shelf calving processes and their contributions to freshwater fluxes into the Amundsen Sea and the Ross Sea.

*Code and data availability*. The source code used for the simulations described here are archived at https://doi.org/10.5281/zenodo.12735787 (Zhang, 2024b). The scripts used to generate the grid and forcing files as well as scripts and data used to generate the figures included in this paper are archived at https://doi.org/ 10.5281/zenodo.14472621 (Zhang, 2024c). The model output can be obtained from the authors upon request. The ERA5 atmospheric reanalysis data, used for atmospheric forcing files, were collected from the Climate Data Store, available at https://cds.climate.copernicus.eu. The GloSea5 reanalysis product, used for the model boundary conditions, were collected from the Copernicus Marine Data Store, available at https://data.marine.copernicus.eu. The BedMachine Antarctica v2 topography data used for the model grid file were collected from the National Snow and Ice Data Center and available at https://nsidc.org/data/nsidc-0756/versions/2.

*Author contributions*. ZZ conceptualized the idea of this work, designed the numerical configurations of the RAISE model and wrote the manuscript draft. CX contributed to the implementation of the model development, numerical simulations and model result validations. CW, YC and XW contributed to the model development. HH performed part of the CTRL simulation. All authors contributed to the writing and editing of this manuscript.

*Competing interests*. The corresponding author has declared that none of the authors has any competing interests.

*Acknowledgements*. This work is funded by Key Research & Development Program of the Ministry of Science and Technology of China (Grant No 2022YFC2807601), the National Natural Science

Foundation of China (Grant No 42476271), the Shanghai Pilot Program for Basic Research of Shanghai Jiao Tong University (Grant No TQ1400201), the Impact and Response of Antarctic Seas to Climate Change (Grant No IRASCC 1-02-01B), and the Shenlan Program funded by Shanghai Jiao Tong University (Grant No SL2020MS021). We thank Michael S. Dinniman from the Old Dominion University for the support on the model development, and Pasquale Castagno and Giorgio Budillon on providing the mooring data from the Italian MORSea project used for model validation, which was financially and logistically supported by the Italian National Programme for Antarctic Research (PNRA).

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
