# Peer review of "The Ross Sea and Amundsen Sea Ice-Sea Model (RAISE v1.0): a high-resolution ocean-sea ice-ice"

_Geoscientific Model Development, 2024_

## Author Comment (AC2)

**Response to Review Comments**

We thank the editor and reviewers for their efforts in making constructive remarks and suggestions, which have significantly improved the quality of our manuscript. Below you can find point-by-point replies to the major and minor comments (*font in Italic*) and the corresponding revisions to the manuscript. In the revised manuscript, revisions are highlighted by light-blue color. We hope that all the editor's and reviewers' concerns have been addressed adequately.

*Reviewer #1:*

*General comments:*

*This study investigated the reproducibility of the sea-ice conditions (sea-ice concentration and production) and oceanic conditions in the Ross Sea using a high-resolution ocean-sea ice-ice shelf model for the Amundsen and Ross Seas, named RAISE v1.0. Additionally, the authors examined the impact of meltwater from the ice shelves in the Amundsen Sea on the water properties in the Ross Sea. Understanding the changes in the coastal water masses around Antarctica is very important due to its significant influence on deep water formation and, subsequently, global thermohaline ocean circulation. However, while reviewing this manuscript, I noticed a significant overlap with the content of the authors' previous publication in JGR-Oceans (Xie et al. 2024, doi:10.1029/2024JC020919). Although there is a slight difference in the model integration period, it is evident that the model used is the same as that in Xie et al. (2024). Moreover, the sensitivity experiments regarding increased meltwater from the Amundsen Sea ice shelves are very similar to each other. Even the description of the model, while arranged differently, appears to be nearly the same. If my understanding is correct, this could be considered a case of duplicate publication. However, if I have misunderstood the extent of the overlap or the novelty of the current work, I would appreciate a clear rebuttal or clarification from the authors. At the very least, it is necessary to properly cite the previous work and clearly highlight the differences.*

We are sorry for giving the reviewer an impression that the model mentioned in this study is the same as that in Xie et al. (2024), which is actually not. Please find our clarifications below.

First, the model developed in this manuscript is an updated version of the one used in Xie et al. (JGR: Oceans, 2024, and also in Zhang et al. (2024) which was published in September of this year). The difference is that surface temperature and salinity are nudged to a monthly mean climatology provided by the World Ocean Atlas 2018 in this model, while Xie et al. (2024) did not apply any nudging. The two versions are developed at the same time. We actually compared the simulations from the two models, and found that the version employing nudging performs better in sea ice production assessed against satellite estimates (Fig. R1 shown below). Additionally, it

better captures the interannual variation of DSW compared with CTD data (Fig. R2). In the manuscript, we did not mention Xie et al. (2024) or Zhang et al. (2024) as they were still under review at the time we submitted this manuscript, and we think it might be inappropriate to cite it without a DOI assigned. In the revised manuscript, we have added such comparisons and emphasized the difference of this model from the one in Xie et al. (2024) and Zhang et al. (2024). In addition, Xie et al. (2024) is focused on scientific problems relevant to the influence of enhanced ice shelf melting in the Amundsen Sea on the Ross Sea water properties, rather than model development and validations. They provided 4 figures for model validation (in the main text and supplementary materials), and while the 3 figures for validating sea ice production, hydrographic variables along the Ross Sea cross-shelf transect and spatial distributions of sea ice concentration are plotted in the same way as those in this manuscript, as mentioned above, these results are based on different model versions and are not duplicate results.

[Figure]

**Fig. R1.** Scatter plots of modelled winter sea ice production versus satellite-estimated winter sea ice production from simulations (a–c) with nudging and (b–d) without nudging for (a and d) the Terra Nova Bay polynya (TNBP), (b and e) the Ross Ice Shelf polynya (RISP), and (c and f) the western RISP.

[Figure]

**Fig. R2.** (a) Time series of summer bottom water salinity near the Ross Island from the model simulation with nudging (i.e. the model version used in this study) and CTD observations (from Jacobs et al. (2022)) during 2003–2019. (b) Time series of summer bottom water salinity near the Ross Island from the model simulation without nudging (the model version used in Xie et al. (2024)) and CTD observations.

Second, as for the sensitivity experiments regarding increased meltwater from the Amundsen Sea ice shelves, the scientific motivations and configurations in Xie et al. (2024) and this work are quite different. Xie et al. (2024) is focused on the impacts of accelerated ice shelf basal melting in the future on DSW formation and CDW intrusion in the Ross Sea, and they increased the melting rates based on future projections of ice shelf melting in the Amundsen Sea from CMIP6 scenarios. In this work, we designed ice shelf sensitivity experiments to address the missing and underrepresented ice shelf melting processes in the model. This adjustment aims to mitigate the overestimation of DSW salinity observed in the model compared to mooring observations. We conducted these experiments by artificially increasing the ice shelf melting rates to match the values estimated from satellite data.

In the revised manuscript, we clarified the differences between the model used in this study and the one used in Xie et al. (2024) and Zhang et al. (2024) (Lines 120–122 and Lines 147–152), and hope in this way we can prevent any potential confusion regarding the originality of our research.

*Specific comments:*

*Figs. 3g, 10, 11*

*I believe it is misleading to claim that the model accurately reproduces the observations simply because the correlation coefficient is significant when seasonal variability is included. Seasonal variability has a strong cyclic pattern, which can lead to a high correlation between the model and observations, even if the model does not truly capture the underlying processes. Evaluating the model's performance without removing the seasonal component can overestimate the model's skill. For a more accurate assessment, the seasonal signal should be removed before calculating the correlation, or the analysis should separately address seasonal and non-seasonal/interannual variability.*

In fact, throughout the manuscript, we avoided using "accurately" to describe the performance of the model, and we mostly used "well" or "reasonably well" for the descriptions. The reviewer is correct that including seasonal cycles normally leads to high correlations between the temporal variability of modelled and observed variables. In Fig. 3g of the revised manuscript, we removed

the seasonal cycle for sea ice concentration (SIC) by subtracting the multi-year climatology from the original SIC values, and the variation of modelled SIC (which is now the anomaly value) is still significantly correlated with the satellite estimate, although the correlation coefficient is reduced (now R=0.68 and P<0.0001).

For Figs. 10 and 11, on the one hand we kept the original plots in the revised version, as if we remove the seasonal cycles, the plots can only show the time series of DSW density anomalies, and in this case we cannot show the effects of improved simulation of the absolute value of DSW density in the Melt+ experiment. On the other hand, we provided plots with seasonal cycles removed (Fig. 10c,d and Fig. 11c,d), and the results show that the temporal variations of modelled and observed DSW neutral density are still significantly correlated. The revised texts are provided in Lines 384–386 and Lines 401–402.

*Fig. 7*

*Regarding the spatial correlation as well, there may still be residual effects from the initial conditions. Even if the model shows a good correlation with observations, it does not necessarily mean that the model accurately reproduces the underlying processes. The initial conditions can strongly influence the spatial patterns, leading to high correlations that may not truly reflect the model's capability to simulate the key dynamics. One could imagine that the spatial correlation between the initial conditions and the observations might yield a similar correlation coefficient. It is important to demonstrate that the higher correlation is due to the high-resolution model resolving fine-scale structures that were not present in the initial conditions, rather than simply reflecting initial condition influence.*

The initial conditions for the model developed in this study come from the simulations produced by a coupled ocean-sea ice-ice shelf model for the Southern Ocean (Dinniman et al., 2015). To verify if the model simulations are strongly affected by the initial conditions, we conducted an additional experiment in which the initial conditions are replaced with the World Ocean Atlas 2018 climatology, and all the other configurations are the same as those in the CTRL simulation. From Figs. R3 and R4 shown below, it can be seen that there are large differences in temperature and salinity between the two initial fields used. Due to computational costs, we only integrated the model in the sensitivity experiment for 5 years from 1998 to 2003, i.e., the spin-up period for the CTRL simulation, and the results show that after 5 years the simulated spatial distributions of temperature and salinity in the sensitivity experiment are very similar to those in CTRL for both the bottom and middle layers (i.e. the DSW and CDW layers, Figs. R5 and R6). These results demonstrate the high spatial correlations between the modelled and observed temperature/salinity in our study are not a result of initial conditions. In the revised version we mentioned that

"Alternative initial conditions from the World Ocean Atlas 2018 (WOA18) are also employed for this model, and we found that after a 5-year spin up period, these conditions yield quite similar model simulations to those initialized by the model results from Dinniman et al. (2015)." (Lines 131–134).

[Figure]

**Fig. R3.** Spatial distributions of temperature from the model simulations using (a and c) original initial conditions and (b and d) the WAO18 data as initial conditions in the (a and b) bottom layer and (c and d) middle layer of the Ross Sea.

[Figure]

**Fig. R4.** Spatial distributions of salinity from the model simulations using (a and c) original initial conditions and (b and d) the WAO18 data as initial conditions in the (a and b) bottom layer and (c and d) middle layer of the Ross Sea.

[Figure]

**Fig. R5.** Spatial distributions of temperature after 5-year spin up period of the model using (a and c) original initial conditions and (b and d) the WAO18 data as initial conditions in the (a and b) bottom layer and (c and d) middle layer of the Ross Sea.

[Figure]

**Fig. R6.** Spatial distributions of salinity after 5-year spin up period of the model using (a and c) original initial conditions and (b and d) the WAO18 data as initial conditions in the (a and b) bottom layer and (c and d) middle layer of the Ross Sea.

*Fig. 4*

*The interannual variability of sea ice production in the Ross polynya, which accounts for a large portion of sea-ice production in the model domain, does not reach the 95% significance level. Therefore, I do not believe the model can accurately reproduce the observed interannual variation in sea-ice production.*

We admit that if we consider the entire Ross Ice Shelf polynya (RISP), the interannual variability of sea ice production from the model is not quite strongly correlated with that from the satellite estimate, which only reaches the 90% confidence level. However, it is demonstrated that DSW primarily exists in the western portion of the RISP and not in the eastern portion (Orsi and Wiederwhol, 2009; Wang et al., 2021), which can also be seen in Fig. 9b. This is because there is more ice shelf meltwater from the Amundsen Sea transported to the eastern portion, and also there is more local meltwater from the Ross Ice Shelf in the eastern portion as the eastern Ross Sea shelf is narrower that facilitates the intrusion of warm CDW to the ice shelf. So sea ice production in the western portion of the RISP contributes most to the DSW production in the RISP. We compared the interannual variability of modelled sea ice production in the western RISP (west of 186°E) to that from the satellite estimate, and the correlation is notably improved (R=0.76, P=0.01). In the revised Fig. 4 both the plots for sea ice production in the entire RISP and the western RISP are provided, and descriptions for ice production in the western RISP are provided in Lines 268–275.

References

Orsi, A. H., and Wiederwohl, C. L.: A recount of Ross Sea water, Deep-Sea Research Part II, 56(13), 778–795, https://doi.org/10.1016/j.dsr2.2008.10.033, 2009.

Wang, X., Zhang, Z., Dinniman, M. S., Uotila, P., Li, X., and Zhou, M.: The response of sea ice and high-salinity shelf water in the Ross Ice Shelf Polynya to cyclonic atmosphere circulations. The Cryosphere, 17, 1107–1126, The Cryosphere, 17, 1107–1126, https://doi.org/10.5194/tc-17-1107-2023, 2023.

*It would be beneficial to include not only a comparison of the water mass properties but also quantitative estimates regarding DSW and AABW formation/production rate in the model.*

While it is important to validate the modelled DSW and AABW production rates, it is difficult to obtain an accurate estimate of these rates from observations, since temporally continuous observations of DSW/AABW for calculating the rates are only available from mooring measurements, which are very sparce in space. Also, hydrographic sensors on these moorings are

only installed at limited depth levels. This makes it difficult to accurately estimate the volume of the DSW and AABW using the mooring observations, and thus difficult to provide a good reference for evaluating the model simulations of the production rates (in unit of $m^3$ $s^{-1}$).

---

## Author Comment (AC3)

**Response to Review Comments**

We thank the editor and reviewers for their efforts in making constructive remarks and suggestions, which have significantly improved the quality of our manuscript. Below you can find point-by-point replies to the major and minor comments (*font in Italic*) and the corresponding revisions to the manuscript. In the revised manuscript, revisions are highlighted by light-blue color. We hope that all the editor's and reviewers' concerns have been addressed adequately.

*Reviewer #2*

*This manuscript presents an Ocean–Sea Ice–Ice Shelf model for Ross Sea and Amundsen Sea (called RAISE). The ocean and sea ice components are primarily based on the ROMS configurations; a static ice shelf is added to the model, which allows inclusion of ice shelf melting. The manuscript presents most technical details for model implementation and validation with available data for key ocean physical processes relevant to production of Dense Shelf Water and Antarctic Bottom Water.*

*Most of the implementation seems straightforward and reasonable in most parts. The simulation spans 17 years (2003-2019) with a prior 5-year spin-up. Numerical experiments with an inert tracer (dye) are run, which show the transport and dispersion patterns of Circumpolar Deep Water (CDW) and ice shelf water (ISW). Another experiment is also run by increasing melting rates of ice shelf from Amundsen Sea ice shelves (Melt+). The results, as compared with available in situ observations or remote sensing, seem reasonable in general with one significant exception (explained below). The general functioning of the system (e.g. transport, mixing, ice production/melting) also appears to be consistent with our knowledge. The main finding is that ice melting from Amundsen Sea plays a key role in determining salinity (and hence density) of dense shelf water (DSW) in this region. It could be worth expanding this topic a bit by explaining, for example, does this affect DSW properties in southwestern Ross Sea shelf? Or does it affect Antarctic Bottom Water (ABW) properties or production?*

We thank the reviewer for the overall constructive comments on this study, and by addressing these issues, we think the manuscript has been significantly improved and several ambiguities have been resolved, resulting in improved clarity. In the revised manuscript, we added plots showing the impacts of increased meltwater from the Amundsen Sea on the salinity or neutral density over the southwestern Ross Sea shelf (Fig. 10, Fig. 12 and Fig. 14), and please see our detailed response to the reviewer's specific comment relevant to this point below. In the original manuscript, we showed that in the experiment with increased meltwater (Melt+), the AABW thicknesses in the open ocean near the Ross Sea can be reduced by over 100 m (Fig. 14b).

*Based on one transect, the model has under-predicted the salinity in the top 150 m by 0.1-0.2 psu (model temperature is slightly, ~1°C, higher than observed) (Figure 6). In contrast, modeled subsurface salinity is higher than observed (by ~0.05 psu). This seems consistent with lower ice concentration than observed (Figure 3). The logic seems to be: Warmer temperature in surface layer leading to less sea ice formation, which in turn leads to insufficient sea brine formation and under-prediction of DSW production. However, this seems contradicting to the sea ice production comparison (Figure 4), which shows model over-predicts the sea ice production. The scatter plot between modeled vs WOD salinity for shelf water also shows a <1 slope, i.e. surface salinity is higher than WOD salinity and vice versa for subsurface water (red dots in Figure 7d)? Perhaps there are spatial mismatch (the transect is on western Ross Sea) or perhaps ice thickness is an issue? In addition, the Melt+ experiment with enhanced ice shelf melting significantly reduces DSW density (salinity), bringing the density much closer to observed. Does this mean now we are getting a smaller slope if we plot a new scatter plot? Regardless, it seems some more clarification is needed to reconcile these.*

The reviewer made a valuable point for explaining the overestimate of sea ice production in the model, while sea ice concentration is underestimated likely associated with the overestimate of sea surface temperature. We then examined the modelled ice thickness against observations from a cruise conducted by the PIPERS (Polynyas, Ice Production and Seasonal Evolution in the Ross Sea) project, which lasted from April to June of 2017 (Ackley et al., 2020). The comparison results (Fig. R7) show that our model does overestimate the sea ice thicknesses in the polynyas and over the Ross Sea shelf, which can contribute to the overestimate of sea ice production. We mentioned this in Lines 259–261 of the revised manuscript, and provided Fig. S1 in the Supplementary Information. Note that the in-situ observations of sea ice are partly from AUV observations and partly from visual ice observations. So there could also be inaccuracies in these data.

[Figure]

**Fig. R7.** (a and c) Sea ice thicknesses from the RAISE model (color shading) and from observations (colored dots) during the PIPERS cruise observations in the (a) Terra Nova Bay and (c) Ross Ice Shelf polynya areas. (b and d) The scatter plot of modelled ice thickness versus observed ice thickness in the cruise in the (b) Terra Nova Bay and (c) Ross Ice Shelf polynya areas. The black solid line indicates the 1:1 ratio line, and the grey dashed line indicates the linear regression fit.

Please note that Fig. 7 is plotted for potential temperature and salinity in the bottom 100-m layer (see the figure caption), so all the points are in the bottom layer. It is true that from Fig. 7d, a slope <1 means the model overestimates salinity in the lower-salinity range and underestimates salinity in the higher-salinity range (high salinities exist mostly over the western Ross Sea shelf). This seems contradictory to the fact that our model overestimates sea ice production, salinity and neutral density in the polynya regions located on the western Ross Sea shelf. In fact, such overestimation can also be seen from Fig. 7c in the Terra Nova Bay polynya and the Ross Ice Shelf polynya, and the reviewer is correct that there are spatial variations in the overestimation or underestimation of salinity over the shelf, even just over the western shelf. We added two sentences to discuss this in the revised manuscript (Lines 318–321). As the reviewer suggested, we made the same plot as Fig. 7 for the Melt+ experiment (Fig. R8 shown below), in fact the slope is slightly increased (from 0.693 to 0.726), and such weak increase might be due to the fact that the meltwater from the Amundsen Sea is mainly carried by slope currents and thus causes larger reduction in salinity in the slope regions (i.e. the lower-salinity range in the scatter plot).

[Figure]

**Fig. R8.** Same as Fig. 7 in the manuscript but for the Melt+ experiment.

*The manuscript can also benefit from some polishing and clarifying a few details. For example, it is unclear, however, how exactly the melting rate under the ice shelf is calculated and applied. It is not clear how Melt+ experiment is being implemented other than that heat/salt transfer coefficients are modulated as done by Nakayama et al. (2020).*

The three-equation parameterization scheme for ice shelf simplifies the thermodynamic processes beneath ice shelves by representing the freezing and melting of sea ice. In the heat conservation equation at the ice shelf/ocean interface, based on thermodynamic equilibrium, the latent heat sink (or source) generated by melting (or freezing) balances the heat loss to the ice and the heat supplied by seawater:

$$Q_I^T - Q_W^T = -\rho_I w_B L_f \qquad (1)$$

Here, $Q_I^T$ and $Q_W^T$ represent the heat fluxes at the ice and seawater interfaces (W m$^{-2}$, positive values indicating upward flux), $\rho_I$ is the ice density (kg m$^{-3}$), $w_B$ denotes the rate of ice melting ($> 0$) or freezing ($< 0$) (m s$^{-1}$), and $L_f$ is the latent heat of fusion of ice (J kg$^{-1}$). The heat flux from the water is generally much greater than that through the ice, and thus the model assumes the ice is perfectly insulating (i.e., $Q_I^T = 0$). Typically, the heat flux from seawater to the ice shelf/ocean interface ($Q_W^T$) is represented using a bulk turbulent transfer formulation:

$$Q_W^T = -\rho_W C_{pw} \gamma_T (T_B - T_W) \qquad (2)$$

Here, $\rho_W$ is the seawater density (kg m$^{-3}$), $C_{pw}$ is the specific heat capacity of seawater (J kg$^{-1}$°C$^{-1}$), $T_B$ is the interface temperature (freezing point), and $T_W$ is the temperature of seawater at a certain distance from the ice shelf/ocean interface. In our model, $T_W$ is defined as the temperature of the uppermost grid cell, a common approach in other studies (Galton-Fenzi et al., 2012; Dansereau et al., 2014). $\gamma_T$ is the heat transfer coefficient (m s$^{-1}$), representing the molecular and turbulent mixing coefficient of heat within the ocean boundary layer adjacent to the ice shelf. Some studies assign a constant value for $\gamma_T$; however, a more commonly parameterization (Timmermann et al., 2012; Holland et al., 2008) is adopted in our model, where $\gamma_T$ is parameterized as a function of the friction velocity (Jenkins et al., 2010). In the model, friction velocity is defined similarly to the frictional drag between the ocean and seabed. The second equation is the salt flux conservation equation. At the ice shelf/ocean interface, the freshwater flux generated by melting or freezing ice with salinity $S_I$ balances the salt flux arriving at the interface from seawater (with the salt flux through the ice shelf, $Q_I^S$, assumed to be zero):

$$-Q_w^S = \rho_I w_B (S_I - S_B) \qquad (3)$$

Here, $Q_w^S$ represents the salt flux at the seawater interface (psu-kg m$^{-2}$ S$^{-1}$), $S_I$ is the ice salinity, and $S_B$ is the salinity at the interface. Sea ice formed from freezing seawater contains some brine, but observations indicate that the salinity content is very low (0.10 or lower), and thus $S_I$ is set to

zero in the model. The salt flux from seawater to the ice shelf/ocean interface, $Q_w^S$, is typically represented in a turbulent diffusive flux form similar to that of heat:

$$Q_w^S = -\rho_W \gamma_S (S_B - S_W) \qquad (4)$$

Here, $\gamma_S$ denotes the salt transfer coefficient (m s⁻¹), and $S_W$ represents the salinity at a certain distance from the ice shelf/ocean interface, which in the model is defined as the salinity of the uppermost grid cell. Due to the differing molecular diffusivities of heat and salt, the values of $\gamma_S$ and $\gamma_T$ differ. However, $\gamma_S$ is also parameterized in our model as a function of the friction velocity. The third equation describes the freezing point of seawater as a weakly nonlinear function of salinity and pressure. By linearizing this relationship, an analytical solution for the coupled system of three equations can be obtained:

$$T_B = aS_B + bP_B + c, \qquad (5)$$

Here, we set the salinity coefficient $a = -5.7 \times 10^{-2}$ °C, the pressure coefficient $b = -7.61 \times 10^{-4}$ °C dbar⁻¹, and in the actual solution, the depth of the ice shelf base is used as a substitute for pressure. The coefficient $c = -9.39 \times 10^{-2}$ °C. By simultaneously solving Equations (1) and (2), as well as Equations (3) and (4), we can obtain the following results:

$$\frac{\rho_I w_B L_f}{c_{pw}} = -\rho_W \gamma_T (T_B - T_W) \qquad (6)$$

$$\rho_I w_B S_B = \rho_W \gamma_S (S_B - S_W) \qquad (7)$$

By combining Equations (5), (6), and (7), we can solve for $S_B$, $T_B$, and the ice shelf melting rate ($w_B$). In the revised manuscript, we provided a brief description of the parameterization scheme for simulating ice shelf melting in Section 2.3 as described above (Lines 183–204).

References

Galton-Fenzi, B. K., Hunter, J. R., Coleman, R., Marsland, S. J., and Warner, R. C.: Modeling the basal melting and marine ice accretion of the Amery Ice Shelf, J. Geophys. Res.-Oceans, 117, C9, https://doi.org/10.1029/2012JC008214, 2012.

Dansereau, V., Heimbach, P., and Losch, M.: Simulation of subice shelf melt rates in a general circulation model: Velocity‐dependent transfer and the role of friction, J. Geophys. Res.-Oceans, 119, 1765-1790, https://doi.org/10.1002/2013JC008846, 2014.

Timmermann, R., Wang, Q., and Hellmer, H. H.: Ice-shelf basal melting in a global finite-element sea-ice/ice-shelf/ocean model, Annals of Glaciology, 53, 60, 303-314, https://doi.org/10.3189/2012AoG60A156, 2012.

Holland, P. R.: A model of tidally dominated ocean processes near ice shelf grounding lines. J. Geophys. Res.-Oceans, 113, C11, https://doi.org/10.1029/2007JC004576, 2008.

Jenkins, A., Nicholls, K. W., and Corr, H. F.: Observation and parameterization of ablation at the base of Ronne Ice Shelf, Antarctica. J. Phys. Oceanogr., 40, 2298-2312, https://doi.org/10.1175/2010JPO4317.1, 2010.

*Figure 10 (TNB mooring) & Figure 12 (Ross Island CTD) suggest modeled salinity was also higher than observed on the western Ross shelf and even areas close to Ross Sea Ice Shelf. Does Melt+ improve salinity simulation in those area too?*

In the revised Fig. 10 and Fig. 12, we added the simulated neutral density or salinity from the Melt+ experiment, and the results show that Melt+ also improved the simulations of density or salinity in these areas. Relevant discussions are added in Lines 470–473 of the revised manuscript.

*Based on dye experiment, it takes 5 years for those dyes released from Amundsen Sea to reach western Ross Sea, suggesting a delay of 5-10 year in the impacts. This may help to explain why salinity improvements at CA1 and CA2 in Melt+ only takes place after mid-2008? Figure 11 did not show salinity changes before 2008, so it is clear if this is truly the case.*

In Fig. 11, we did not show the neutral density (I guess the reviewer mean this variable instead of salinity) changes before 2008, because the mooring data are only available from 2008 to 2016. In fact, if we look carefully, salinity had already been notably decreased in Melt+ in early 2008 (January to March) compared to CTRL, while in mid 2008 it became close to the value in CTRL again. Another issue that we need to clarify is, in Fig. 11 we just showed the distributions of ISW dyes from the Amundsen Sea ice shelves 5 years after their release time (as mentioned in the figure caption), and 5 years is not transport time for the ISW to be carried from the Amundsen Sea to the Ross Sea. In fact, the transport time is shorter, which is approximately 2 years. We calculated the difference in salinity from Melt+ and CTRL at bottom depths of CA1 and CA2 from 2003 (the start time of Melt+) to 2013 (Fig. R9). We can see that from 2003 to 2005, the salinity reduction in Melt+ compared to CTRL is not significant, confirming the fact it takes about 2 years for increased meltwater in the Amundsen Sea to influence salinity in the Ross Sea. We added such information in Lines 345–346 in the revised manuscript.

[Figure]

**Fig. R9.** Time series of differences in salinity from the Melt+ and CTRL simulations (Melt+ minus CTRL) during 2003–2013 at (a) 1735 m at the CA1 location and (b) 1929 m at the CA2 location.

*It seems higher model SIP than observed (Figure 4) is contradictory to the lower ice concentration than observed (Figure 3). But this may also mean the model over-estimates the ice thickness. So perhaps some clarifications about how model performs on sea ice thickness will be helpful.*

Please see our response to the review's earlier comments about this issue, and we provided ice thickness validation results in the Supplementary Information.

*Some editorial comments/suggestions*

*Page 2, line 31-34, Is this true there were no modeling studies examining this issue?*

Up until now we have not noticed studies that evaluated modelled DSW variations in the Southern Ocean on different timescales (interannual and long-term variability as in this study) as well as their driving forcings. But in case that we may have missed any studies, we revised this sentence to "This study represents an attempt to thoroughly evaluate the DSW properties and associated ocean-sea ice-ice shelf coupling processes among modelling studies in the Southern Ocean, using …" (Lines 31–33).

*Page 2, line 35, suggest changing "DSW" -> "DSW properties"?*

Revised accordingly (Line 35).

*Page 2, line 36, "which are not seen in DSW studies before". It is unclear what have not been seen, temporal variations or observations?*

The original sentence is "In particular, the modelled temporal variations of DSW properties in polynyas and its key export passages are compared with long-term mooring observations, which are not seen in DSW studies before". We changed it to "In particular, the modelled temporal variations of DSW properties in polynyas and its key export passages are compared with long-term mooring observations, which are rarely seen in studies of the DSW temporal variability before" (Lines 35–37).

*Page 3, line 41, "under-estimate" -> "an under-estimate"*

Revised accordingly (Line 41).

*Page 5, figure 1 lower panel. It is worth noting the model grid is not conforming to orthogonality in those two southern corners. This may be intentional and shall not affect model results since they are land points.*

Our intention is that as far as the model can cover the entire Ross Ice Shelf, the smaller model domain (and thus less grid points) the better. This does produce some model grid points at the southwestern and southeastern corners not conforming to orthogonality very well. But as the reviewer mentioned, there are land points and shall not affect the model results.

*Page 6, entire paragraph, so some aspects of the interactions between ice shelf melting and DSW may have been examined.*

It is true that these studies discussed some aspects of the interactions between ice shelf melting and DSW, which are also mentioned in this paragraph. But as pointed out in this paragraph, the models used in these studies just cover the Ross Sea, and the effects of ice shelf meltwater from the Amundsen Sea are artificially simulated by changing salinity at the open boundaries of the models. Such operations would to some extent help reveal the effects of meltwater on the Ross Sea water properties, but cannot provide an accurate estimate of such effects.

*Page 7, line 117, "finite-volume" should be "finite-difference"*

Thanks for raising this critical, yet long debated topic. ROMS has been described as a finite-difference model in many literatures over the years, which is correct since the primitive equations are discretized with the finite-difference method. However, a commentary article by Shchepetkin and McWilliams (2009), who are early developers of ROMS, pointed out that ROMS is technically a finite-volume model since the tracer equations are discretized in volume integrated form. Therefore, unlike other finite-difference models, tracer conservation is always guaranteed. The same argument is brought up by Shchepetkin in this post on ROMS forum (https://www.myroms.org/forum/viewtopic.php?t=106), which states that "If one wants to guarantee things like integral conservation of something, and the grid is curvilinear, then "finite volume" is basically the only way to go." He argued that for structured grid models, the key of 'finite volume' is conservation of volume and tracers, and since ROMS's dynamic kernel guarantees conservation, it should be classified as a 'finite volume model'. In this manuscript, we adopted Shchepetkin and McWilliams (2009)'s view, and thus refers ROMS as a finite-volume model. We are aware of the disputes in this community but decide to leave the statement in this manuscript to maintain the integrity of the description.

Reference

Shchepetkin, A., and McWilliams, J.: Correction and commentary for "ocean forecasting in terrain-following coordinate: Formulation and skill assessment of the regional ocean modeling system" by Haidvogel et al., 3595–3634, Journal of Computational Physics, 228, 8985–9000, 2009.

*Page, 8, line 133, maybe spell out "EN4" and add either a link or a reference?*

The reference for EN4 is provided in the revised version (Line 142). In all references and webpages describing EN4, only the abbreviation is used and we were not able to find a full name for it. So we just used EN4 here.

*Page 8, line 135, "Below 1000 m (the isobath at the shelfbreak)". Is 500m a better delineator for shelf-break. 1000 m may be more like upper slope.*

The 1000-m isobath is widely used as a delineator for the Antarctic shelf break in previous studies, including studies for the Ross Sea (e.g., Moorman et al. 2020; Morrison et al. 2020; Silvano et al. 2020). Following Goddard et al. (2017), the 1000-m isobath closely tracks the Antarctic Slope Current (ASC) and the Antarctic Slope Front (ASF) (Huneke et al. 2022), making it suitable for being used as a boundary between the continental shelf and the open ocean. In addition, its

advantages lie in providing a consistent boundary for circumpolar analysis, accounting for the variable bathymetry along the Antarctic continental margin.

References

Huneke, W. G. C., Morrison, A. K., and Hogg, A. M.: Spatial and Subannual Variability of the Antarctic Slope Current in an Eddying Ocean–Sea Ice Model, Journal of Physical Oceanography, 52, 347-361, https://doi.org/10.1175/JPO-D-21-0143.1, 2022.

Moorman, R., Morrison, A. K., and McC. Hogg, A.: Thermal Responses to Antarctic Ice Shelf Melt in an Eddy-Rich Global Ocean–Sea Ice Model, Journal of Climate, 33, 6599–6620, 10.1175/JCLI-D-19-0846.1, 2020.
Morrison, A., Hogg, A., England, M., and Spence, P.: Warm Circumpolar Deep Water transport toward Antarctica driven by local dense water export in canyons, Science Advances, 6, eaav2516, 10.1126/sciadv.aav2516, 2020.

Silvano, A., Foppert, A., Rintoul, S. R., Holland, P. R., Tamura, T., Kimura, N., Castagno, P., Falco, P., Budillon, G., Haumann, F. A., Naveira Garabato, A. C., and Macdonald, A. M.: Recent recovery of Antarctic Bottom Water formation in the Ross Sea driven by climate anomalies, Nature Geoscience, 13, 780-786, 10.1038/s41561-020-00655-3, 2020.

*Page 10, line 153, so does the release only take place once, or is it released continuously? If the latter, for how long?*

The dyes are released continuously released during the simulation periods of the experiments. This is clarified in the revised version (Lines 158–159).

*Page 10-11, section 2.3, how is ice shelf melting simulated? Since this is one of most critical factors in this study, perhaps spell out some details how this is being implemented?*

Please see our response to the Reviewer's comment above.

*Page 11, line 179, Melt+, increasing ice melt rate by how much?*

The value is actually provided in Section 4.5 (~450 Gt yr$^{-1}$) when we describe the experiment in more details, so we added "(see the details in Section 4.5)" here (Line 204).

*Page 11, line 187, add WOD behind "World Ocean Database"*

"WOD" is added (Line 211).

*Page 12, line 197. Add period behind 2016.*

We apologize for this typo, and a period is added behind 2016.

*Page 12, line 205, define SIC here*

"SIC" is replaced with sea ice concentration here (Line 229).

*Page 13, line 216, remove "sea ice concentration" now*

Revised accordingly (Line 241).

*Page 13, section 4.1, how about sea ice thickness, any data to compare?*

Please see our response above.

*Page 14, line 234-240. Not sure about this – so I assume satellite estimates of sea ice production are based on observed (perhaps also estimated) ice thickness and ice concentration? But does this statement suggest that these estimates did not take into accounts the change in ice thickness due to the bottom portion of sea ice?*

In fact, satellite estimates of sea ice production are based on estimated ice thickness derived from atmospheric heat fluxes, which are normally from reanalysis products (e.g., Tamura et al., 2008; 2016; Nihashi and Ohshima, 2015; Nihashi et al., 2017). Nihashi et al. (2024) demonstrates notable differences between the satellite estimates of ice thickness using ERA5 and ERA-Interim. Inaccuracy in heat fluxes data can affect both the estimates of sea ice thickness and production rates. In addition, as acknowledged in these studies, oceanic heat fluxes are not included in the

estimate of sea ice thickness and ice production, which will affect ice growth at the bottom of sea ice and is important source for the estimation errors.

References

Nihashi, S. & Ohshima, K. I. Circumpolar Mapping of Antarctic Coastal Polynyas and Landfast Sea Ice: Relationship and Variability. Journal of Climate, 28, (2015).

Nihashi, S., Ohshima, K. I. & Tamura, T. Sea-Ice Production in Antarctic Coastal Polynyas Estimated From AMSR2 Data and Its Validation Using AMSR-E and SSM/I-SSMIS Data. IEEE Journal of Selected Topics in Applied Earth Observations and Remote Sensing 10, (2017).

Nihashi, S., Ohshima, K. I. & Tamura, T. Reconstruct the AMSR-E/2 thin ice thickness algorithm to create a long-term time series of sea-ice production in Antarctic coastal polynyas. Polar Science (2024).

Tamura, T., Ohshima, K. I. & Nihashi, S. Mapping of sea ice production for Antarctic coastal polynyas. Geophysical Research Letters 35, 2007GL032903 (2008).

*Page 16, line 262, add "underestimates" to read like "overestimates temperature and underestimates salinity in the surface layer…". Also modeled subsurface temperature on top of the bank appears to warmer than observed. Overall, the model predicts a deeper surface mixed layer.*

Following the reviewer's suggestion, this sentence has been revised to read as "Compared to observations, the model slightly overestimates temperature and underestimates salinity in the surface layer. In the subsurface layer, the model has lower temperature in the open ocean and higher temperature on the shelf, indicating stronger CDW intrusion in the model relative to the observational data; the subsurface salinity is underestimated in the model." (Lines 299–302).

*Page 19, line 293, suggest change "with temperature" to "where water temperature is above 0ºC"? And after this, maybe start another sentence, sth like "As seen in Fig. 8a …"?*

This sentence is changed to "in the open ocean where water temperature is above 0°C. As seen in Fig. 8a that shows the CDW dye values …" (Line 332).

*Page 19, line 298, suggest deleting "the role of"*

Revised accordingly, and now the sentence reads as "which could result from a southward flow …" (Line 337).

*Page 19, line 301-303, this sentence is confusing. Maybe revise this to sth. like, "Compared to the western portion of the RIS, there is more ISW beneath the eastern portion, indicating stronger influence on the hydrography over the Ross Sea shelf.*

We admit that is this sentence is ambiguous, and following the reviewer's suggestion, we revised it to read as "Compared to the western portion of the RIS, there is more ISW beneath the eastern portion, indicating stronger influence on the hydrography over the Ross Sea shelf" (Lines 340–342).

*Page 20, line 312, I assume here dye concentration (and Figure 9) is from bottom layer?*

The dye values in Fig. 9 are vertically integrated results. We added this information in Fig. 9 caption.

*Page 21, line 321-322, do tides play any role in spreading dyes?*

Tidal processes may facilitate the spreading of DSW in the RISP region, particularly at the ice shelf front. As reported by Arzeno et al. (2014), data from two moorings deployed approximately 6 and 16 km south of the ice front east of Ross Island indicate that about half of the variance in the currents there are attributed to tidal influences, predominantly from diurnal components, while the remainder is due to subtidal oscillations with periods of a few days. Concurrently, Jendersie et al. (2018) proposed that the presence of a buoyancy-driven cyclonic circulation (shown in Figure 3b of their work) near the Ross Ice Shelf (RIS) plays a significant role in the southward spreading of DSW into the ice shelf cavity west of 180°. Their sensitivity experiments showed that RIS cavity temperatures and melting rates are very similar between tidal and non-tidal simulations, which indicates that the distributions of DSW should also be similar between these experiments as DSW can contribute to ice shelf melting. Together, these processes highlight the complex dynamics at play in the region. Comprehensive analysis on the role of tides in the DSW spreading beneath the RIS could be undertaken in future work using the RAISE model. As tides may play a role, we revised the sentence to "which can be associated with the southward flow as mentioned above as well as the role of tidal currents (Arzeno et al., 2014)" (Line 361).

Arzeno, I. B., Beardsley, R. C., Limeburner, R., Owens, B., Padman, L., Springer, S. R., Stewart, C. L., and Williams, M. J. M.: Ocean variability contributing to basal melt rate near the ice front of Ross Ice Shelf, Antarctica, Journal of Geophysical Research: Oceans, 119, 4214-4233, https://doi.org/10.1002/2014JC009792, 2014.

Jendersie, S., Williams, M. J. M., Langhorne, P. J., and Robertson, R.: The Density-Driven Winter Intensification of the Ross Sea Circulation, Journal of Geophysical Research: Oceans, 123, 7702–7724, 10.1029/2018JC013965, 2018.

*Page 22, line 333, "in both middle and bottom" -> "in both layers"*

"in both the middle and bottom layers" is changed to "in both layers" (Line 380).

*Page 22, line 335, Section 4.1?*

It is actually Section 4.4, and we revised this sentence to "see the discussions in Section 4.4" (Line 382).

*Page 22, Figure 10 caption, "... (b) Same as (a) but density"?*

Sorry for this typo, and "Same as (a) but for salinity" in the original sentence has been changed to "Same as (a) but for neutral density".

*Page 23, line 345, again, this should add "properties" so it reads like "... variations of DSW properties…"*

This sentence is revised to "The model also performs well in simulating the temporal variations of DSW properties …" (Line 388).

*Page 25, line 384, suggest changing "falls in the range of" to "is in line with" or "is on the upper end of"*

"falls in the range of" is replaced with "is in line with" (Line).

*Page 25, line 388, "the satellite estimates" - > "all the satellite estimates"*

Revised accordingly (Line 435).

*Page 29, line 455, again it seems higher model SIP is contradictory to the less ice concentration*

Please see our response to the reviewer's comment above.

---

## Author Response (AR2)

**Response to Review Comments**

We thank the editor and reviewers for their efforts in making constructive remarks on our revised manuscript. Below you can find point-by-point replies to minor comments from Referee #1 (*font in gray and Italic*) and the corresponding revisions to the manuscript. In the revised manuscript ("Tracked-changes" version), revisions are highlighted by blue color. We hope that all the reviewer's concerns have been addressed adequately.

**Referee #1:**

*I appreciate the authors' efforts in addressing my previous comments in the revised manuscript. I believe the manuscript is now close to being suitable for publication. I would appreciate it if the authors could consider the following minor points for further improvement.*

We thank the reviewer for the overall constructive comments on the revised manuscript, and please find our pointwise response to the comments below.

*Minor comments:*

*Introduction L 67: Another paper also conducted numerical experiments to examine the response in the Ross Sea to increased meltwater in the Amundsen Sea.*
*Kusahara and Hasumi (2013) "Modeling Antarctic ice shelf responses to future climate changes and impacts on the ocean", JGR-Oceans, doi:10.1002/jgrc.20166*

We thank the reviewer for suggesting this reference relevant to our study, and we have added it in the revised manuscript (Line 67).

*Section 2.3: The values of the parameters are missing. Please consider adding a table to specify the values for $\rho\_i$, $\rho\_w$, $C\_pw$, $L\_f$, $\gamma\_t$, and $\gamma\_s$. Additionally, could you clarify the type of formulation used for $\gamma\_t$ and $\gamma\_s$?*

We added the values for $\rho_I$, $\rho_W$, $L_f$ and $C_{pw}$ in the text explaining Equation (1) (Lines 186–187 in the revised manuscript). For the transfer coefficients $\gamma_T$ and $\gamma_S$, we mentioned that "$\gamma_T$ and $\gamma_S$ are specified following McPhee et al. (1987) that assumes a viscous molecular sub-layer adjacent to the ice–ocean boundary." (Lines 195–196).

*Quantitative Assessment of DSW/AABW: In my previous comment, I suggested a quantitative assessment of DSW/AABW. The authors responded that this was not addressed due to the difficulty in observation. However, considering that the model in this manuscript has been thoroughly validated with observation-based data, I still believe it would be meaningful to include model-based estimates in the manuscript.*

We agree that adding quantitative estimates for the DSW/AABW production/transport will provide useful information for the scientific community. In addition, recently we found a few literatures providing rough estimates of the DSW (or HSSW) production rate in the Terra Nova Bay polynya (TNBP) , based on limited mooring observations or model simulations. In the revised manuscript, we computed the annual average production rate of DSW in the TNBP using the RAISE model simulation from 2003 to 2019, and compared our result with estimates from earlier studies. Such comparison results are added in Lines 391–395 of the revised version as "The annual average DSW production rate in the TNBP estimated from the RAISE model simulation is 0.33 Sv, which is between the estimate of 0.28 Sv from Jendersie et al. (2018) using a coupled ocean-sea ice model for the Ross Sea and the estimate of 0.43 Sv from Miller et al. (2024) using observations from a mooring in the TNBP. The estimated annual average DSW production rate in the RISP from the model is 1.23 Sv.". In addition, we added a figure showing the time series of DSW transports at the slope through the major three passages (troughs, Figure 12 in the revised manuscript), and derived the annual mean transport across each trough. We added these information in Lines 402–407 of the revised manuscript.

References

McPhee, M. G., Maykut, G. A., & Morison, J. H.: Dynamics and thermodynamics of the ice/upper ocean system in the marginal ice zone of the Greenland Sea, Journal of Geophysical Research: Oceans, 92, 7017-7031, https://doi.org/10.1029/JC092iC07p07017,1987.

Jendersie, S., Williams, M. J. M., Langhorne, P. J., and Robertson, R.: The Density-Driven Winter Intensification of the Ross Sea Circulation, Journal of Geophysical Research: Oceans, 123, 7702–7724, 10.1029/2018JC013965, 2018.

Miller, U. K., Zappa, C. J., Gordon, A. L., Yoon, S. T., Stevens, C., & Lee, W. S.: High Salinity Shelf Water production rates in Terra Nova Bay, Ross Sea from high-resolution salinity observations, Nature Communications, 15, 373, https://doi.org/10.1038/s41467-023-43880-1, 2024.